



# Evaluation of the accuracy of thermal dissociation CRDS and LIF techniques for atmospheric measurement of reactive nitrogen species

Caroline C. Womack[1,2], J. Andrew Neuman[1,2], Patrick R. Veres[1,2], Scott J. Eilerman[1,2], Charles A. Brock[1], Zachary C. J. Decker[3], Kyle J. Zarzana[1,2], William P. Dube[1,2], Robert J. Wild[1,2], Paul J. Wooldridge[4], Ronald C. Cohen[4,5], Steven S. Brown[1,3]

[1]Chemical Sciences Division, Earth Science Research Laboratory, National Oceanic and Atmospheric Administration, Boulder, CO 80305 USA
[2]Cooperative Institute for Research in Environmental Sciences, University of Colorado, Boulder, CO 80309 USA
[3]Department of Chemistry and Biochemistry, University of Colorado, Boulder, CO 80309 USA
[4]Department of Chemistry, University of California, Berkeley, CA 94720 USA
[5]Department of Earth and Planetary Science, University of California, Berkeley, CA 94720USA

*Correspondence to*: S. S. Brown (steven.s.brown@noaa.gov)

**Abstract.** The sum of all reactive nitrogen species ($NO_y$) includes $NO_x$ ($NO_2$ + NO) and all of its oxidized forms, and the accurate detection of $NO_y$ is critical to understanding atmospheric nitrogen chemistry. Thermal dissociation (TD) inlets, which convert $NO_y$ to $NO_2$ followed by $NO_2$ detection, are frequently used in conjunction with techniques such as laser induced fluorescence (LIF) and cavity ringdown spectroscopy (CRDS) to measure total $NO_y$ when set at >600 °C, or speciated $NO_y$ when set at intermediate temperatures. We report the conversion efficiency of known amounts of several representative $NO_y$ species to $NO_2$ in our TD-CRDS instrument, under a variety of experimental conditions. We find that the conversion efficiency of $HNO_3$ is highly sensitive to the flow rate and the residence time through the TD inlet, as well as the presence of other species that may be present during ambient sampling, such as ozone ($O_3$). Conversion of $HNO_3$ at 400 °C, nominally the set point used to selectively convert organic nitrates, can range from 2-6% and may represent an interference in measurement of organic nitrates under some conditions. The conversion efficiency is strongly dependent on the operating characteristics of individual quartz ovens, and should be well calibrated prior to use in field sampling. We demonstrate quantitative conversion of both gas phase $N_2O_5$ and particulate ammonium nitrate in the TD inlet at 650 °C, the temperature normally used for conversion of $HNO_3$. $N_2O_5$ has two thermal dissociation steps, one at low temperature representing dissociation to $NO_2$ and $NO_3$, and one at high temperature representing dissociation of $NO_3$, which produces exclusively $NO_2$ and not NO. We also find a significant interference from partial conversion (5-10%) of $NH_3$ to NO at 650 °C in the presence of representative (50 ppbv) levels of $O_3$ in dry zero air. Although this interference appears to be suppressed when sampling ambient air, we nevertheless recommend regular characterization of this





interference using standard additions of $NH_3$ to TD instruments that convert reactive nitrogen to NO or $NO_2$.

## 1 Introduction

The catalytic cycling of nitrogen oxides ($NO_x = NO + NO_2$) plays a key role in the formation of
tropospheric ozone ($O_3$) from the photooxidation of VOCs. Reactive nitrogen species, such as alkyl and
multifunctional nitrates (ANs, $RONO_2$), peroxy nitrates (PNs, $RO_2NO_2$) and nitric acid ($HNO_3$) serve as
reservoirs and sinks of $NO_x$. The formation of these species results in chain termination that determines
the efficiency of the $O_3$ production cycle, and can also transport $NO_x$ far from the original emission
source. For this reason, total reactive nitrogen ($NO_y = NO + NO_2 + RONO_2 + RO_2NO_2 + HNO_3 + HONO$
$+ NO_3 + 2 \times N_2O_5 +$ aerosol nitrates) is an important tracer in monitoring tropospheric $O_3$ production. Its
accurate detection is critical in field measurements of ambient air quality, as $O_3$ is a known health risk,
and a number of regions across the US are currently in non-attainment or near non-attainment with
national ambient air quality $O_3$ standards (EPA, 2016). However, the sources and fates of $NO_y$ species are
complex and remain poorly characterized in some regions. Measured total reactive nitrogen has in some
cases deviated significantly from the sum of the measured individual components, $\Sigma NO_{y,i}$ (see (Fahey et
al., 1986;Bradshaw et al., 1998;Neuman et al., 2012) and others referenced within). This unmeasured
$NO_y$, sometimes referred to as "missing $NO_y$", indicates the need for a more complete understanding of
total and speciated reactive nitrogen, and for accurate analytical instrumentation for $NO_y$ measurement
(Crosley, 1996;Williams et al., 1998;Day et al., 2003).

20          Techniques that detect the major individual components of $NO_y$ include detection of NO and $NO_2$
by chemiluminescence (Ridley and Howlett, 1974;Kley and McFarland, 1980), cavity ringdown
spectroscopy (CRDS) (Fuchs et al., 2009), or laser induced fluorescence (Thornton et al., 2000), and
detection of $HNO_3$ by chemical ionization mass spectrometry (CIMS) (Fehsenfeld et al., 1998;Huey et al.,
1998;Neuman et al., 2002;Huey, 2007) or mist chamber sampling (Talbot et al., 1990). Additionally,
speciated peroxy acetyl nitrates (PANs) have been detected by gas-chromatography electron capture
detection (Darley et al., 1963;Flocke et al., 2005) and CIMS (Slusher et al., 2004), while $N_2O_5$ and $ClNO_2$
have been detected by CRDS (Dubé et al., 2006;Osthoff et al., 2008) and CIMS (Kercher et al., 2009).
However, fewer methods have been developed for detection of the broad suite of individual alkyl and
multifunctional nitrates, which have been suggested to comprise upwards of 20% of $NO_y$ in the
midlatitude continental boundary layer and may be higher in remote locations (O'Brien et al., 1995;Day et
al., 2003;Beaver et al., 2012;Xiong et al., 2015;Lee et al., 2016). An alternative to detecting individual
components of $NO_y$ is the use of a molybdenum oxide or gold catalyst in the presence of CO to reduce all





$NO_y$ species to NO, followed by NO detection by chemiluminescence (Winer et al., 1974;Fahey et al., 1986), though catalyst-based techniques are known to require frequent cleaning, and are potentially sensitive to contamination and to interferences at ambient levels of ammonia, HCN, acetonitrile, and R-$NO_2$ compounds (Crosley, 1996;Kliner et al., 1997;Bradshaw et al., 1998;Williams et al., 1998;Day et al., 2002). An alternative method developed by Day and co-workers (Day et al., 2002) uses a quartz thermal dissociation (TD) inlet to rapidly thermally convert nearly all $NO_y$ species to $NO_2$, which is then detected by laser induced fluorescence (LIF). The $NO_y$ species in the TD inlet undergo the following reaction

$$XNO_2 + heat \rightarrow X + NO_2 \tag{1}$$

where X is HO, RO, or $RO_2$. Heated inlets had previously been used to dissociate PNs (Nikitas et al., 1997), but the TD inlet developed by Day et al. (2002) takes advantage of the different O-N bond energies of ANs, PNs, and nitric acid to separately and selectively detect these three classes of $NO_y$. A plot of measured $NO_2$ signal as a function of inlet temperature (hereafter referred to as a "thermogram") yields a stepwise dissociation curve with increases in signal near 100, 300 and 500 °C, corresponding to the dissociation of PNs, ANs, and $HNO_3$ respectively. By setting the TD oven temperature to one of the three plateaus, they were able to measure each class of $NO_y$, by comparison of the $NO_2$ signal in a given channel to the signal measured at the adjacent lower temperature plateau.

In recent years, a suite of other instruments have incorporated this $NO_y$ TD inlet method into existing techniques that measure $NO_2$ or the radical cofragment X in Eq. (1), such as chemical ionization mass spectrometry (TD-CIMS) (Slusher et al., 2004;Zheng et al., 2011;Phillips et al., 2013), cavity ringdown spectroscopy (TD-CRDS) (Paul et al., 2009;Thieser et al., 2016), and cavity attenuated phase shift spectroscopy (TD-CAPS) (Sadanaga et al., 2016). Each instrument has its own advantages and disadvantages. For example, TD-LIF detects $NO_2$ at low pressure following thermal dissociation, which minimizes secondary recombination reactions of the dissociated radicals, but is subject to interferences from ambient levels of NO and $NO_2$ (Paul et al., 2009;Wooldridge et al., 2010). TD-CIMS can differentiate between the different types of PNs or ANs, but requires regular calibration of each species, not all of which have native standards readily available. TD-CAPS is subject to interferences from glyoxal and methylglyoxal (Sadanaga et al., 2016). TD-CRDS is an absolute measurement, but can be subject to other interferences, as discussed in Sect. 3.

Recent TD inlet studies (Day et al., 2002;Paul et al., 2009;Thieser et al., 2016) have measured the conversion efficiency for several AN and PN species with known concentrations in a laboratory setting. These studies all note the possibility of secondary reactions that either increase or decrease the $NO_2$ signal. For example, recombination reactions to reform the AN or PN species prior to reaching the



detector will result in a negative artifact in $NO_2$ (too little $NO_2$ measured). Likewise, ambient levels of $O_3$ in the sampled air (or the O atoms that form in $O_3$ pyrolysis) may react in the oven with NO to form $NO_2$, resulting in a positive artifact (Pérez et al., 2007), though this reaction rate depends on the TD inlet pressure and flow rate (Wooldridge et al., 2010). Day et al. (2002) found that recombination reactions

were significant for PNs, but caused minimal problems for nitric acid, since the OH radical is far more likely to be lost to the walls of the oven than to recombine with $NO_2$. More significant is the reaction of dissociated $RO_2$ and $HO_2$ radicals with ambient levels of NO and $NO_2$. Thieser et al. (2016) parameterized the bias in peroxyacetyl nitrate and 2-propyl nitrate detection in their inlet as a function of ambient NO and $NO_2$ concentrations, but noted that these parameterizations may vary for other PNs or

ANs. These effects are generally considered minor compared to other uncertainties in the measurement, but in cases where the concentration of one category of $NO_y$ species far exceeds the others, such as the high $HNO_3$:ANs ratios in Pusede et al. (2016), speciated measurements can be significantly affected by biases in measurements of the other $NO_y$ compounds.

     A four-channel CRDS instrument (hereby referred to as the NOAA TD-CRDS instrument) for

detection of nitrogen oxides was recently developed (Wild et al., 2014). In this instrument, one channel is equipped with a TD inlet set at 650 °C and is used to measure all $NO_y$ species (including $NO_2$, and NO by chemical conversion with an $O_3$ addition to $NO_2$). Two other channels simultaneously monitor $NO_2$ and NO, and so a measurement of $NO_z$ (= $NO_y$ – $NO_x$) can be derived. Because NO is intentionally detected as $NO_2$ in the $NO_y$ channel, this instrument avoids the majority of the NO ↔ $NO_2$ interconversion

interferences that affect many other thermal dissociation instruments. Analogous to the studies which measured the conversion efficiencies of ANs and PNs (Day et al., 2002;Paul et al., 2009;Sadanaga et al., 2016;Thieser et al., 2016), we present here an analysis of the conversion efficiencies of several other $NO_y$ species, and the interferences that affect the operation of this high temperature inlet. These interferences include the temperature dependence of $HNO_3$ conversion, which is important to understanding both its

quantitative conversion at 650°C as well as its potential to interfere with measurements of ANs at lower temperatures. We also compare these results to those from the TD-LIF instrument of Day et al. (2002), hereby referred to as the Berkeley TD-LIF instrument. Additionally, we report the temperature dependence of $N_2O_5$ conversion, which is shown to occur in two steps, the conversion efficiency of ammonium nitrate aerosol, and finally the interference of $NH_3$ through its partial conversion to NO.



## 2 Methods

### 2.1 Thermal dissociation cavity ringdown spectroscopy (TD-CRDS)

Cavity ringdown spectroscopy is a direct absorption technique for measuring the concentration of trace gases (Fuchs et al., 2009). The four-channel 405 nm NOAA TD-CRDS instrument, which has been used by our group in both lab-based studies and atmospheric sampling (Wild et al., 2014;Wild et al., 2016), simultaneously measures ambient $NO_2$ in one channel, while chemically converting NO and $O_3$ to $NO_2$ in the second and third channels, and thermally converting $NO_y$ to $NO_2$ in a TD oven in the fourth channel. In this study, we have used only the $NO_y$ channel to study the conversion efficiency of several reactive nitrogen species to $NO_2$. Figure 1 shows a schematic of the relevant instrument plumbing and optical cavity. The details of the optical cavity can be found in (Wild et al., 2014); only a brief description of the optical system and the details of the TD inlet that deviate from that study will be described here.

Sampled air is pulled into a 50 cm long high-finesse optical cavity capped by highly reflective end mirrors, with purge flows of 25 sccm (cubic centimeters per minute at 273.15 K and 1 atm) added in front of each mirror to maintain mirror cleanliness. The output of a 0.5 nm bandwidth, continuous wave diode laser centered at approximately 405 nm and modulated at 2 kHz is passively coupled into one end of the optical cavity. The laser light builds up in the cavity, and when it is modulated off, the decaying output light intensity is monitored by photomultiplier tube on the far side of the cavity. The measured light decays are summed and fit at a 1 Hz repetition rate to yield the ringdown time $\tau$. The ringdown time is inversely related to the concentration of the absorbing gas, $NO_2$ in this case, which can be derived as

$$[NO_2] = \frac{R_L}{c\sigma}\left(\frac{1}{\tau} - \frac{1}{\tau_0}\right) \qquad (2)$$

where $R_L$ is the ratio of $d$, the mirror separation length, and $l$, the distance over which the sample is present. The speed of light is represented by $c$, $\sigma$ is the absorption cross section of $NO_2$, and $\tau_0$ is the ringdown time of a reference cavity without any absorbing gases, which is obtained by flushing the cavity with an excess flow of zero air for 30 seconds every 10 to 20 minutes. If purge volumes were not used, the $R_L$ term in Eq. (2) would simply be 1, but since purge volumes are used here, $\sigma/R_L$ is calibrated regularly by filling the cavity with several different known $NO_2$ concentrations and calculating the slope of the measured optical extinction vs $[NO_2]$ as described in Washenfelder et al. (2011). This value was measured approximately once per month during laboratory tests with this instrument, but was constant to within ±1%, with an average value of $6.25 \times 10^{-19}$ cm$^2$. More regular calibrations of the $\sigma/R_L$ value during recent field studies show similar stability. The $NO_2$ signal can be measured with a lower detection of 18 pptv (1$\sigma$) in 1 second (Wild et al., 2014).



The $NO_y$ TD oven inlet consists of a quartz tube (0.39 cm ID, 63 cm in length, 38 cm of which is heated) wrapped in Nichrome wire and insulated with fiberglass. The flow rate through the inlet and optical cavity is controlled by a mass flow controller on the downstream side of the optical cavity. Because the standard flow rate is held constant during each experiment, the volumetric flow rate, and

therefore the TD residence time, varies with oven temperature. For example, the 4.5 $cm^3$ inner volume of the oven results in an oven residence time of 30 - 100 ms at a flow rate of 1.9 slpm (slpm = liters per minute at 273.15 K and 1 atm) for temperatures from 25 – 650 °C. 1.9 slpm represents the normal operating conditions of this instrument, but flow rates between 0.25 and 3 slpm were tested, which provides oven residence times between 20 and 400 ms. The temperature of the TD oven is monitored by a

thermocouple mounted to the outer side of the quartz tube, and therefore is slightly lower than the temperature of the gas. However, inserting a temperature probe into the inner part of the TD inlet yields a temperature profile, shown in Fig. S1, which approaches the temperature set point by the end of the inlet. All oven temperatures described hereafter refer to the measured thermocouple temperature. After passing through the TD oven, the gas cools to room temperature in the non-heated portion of the quartz tube,

passes through a particle filter (47 mm diameter, 1 µm pore size PTFE membrane) to remove non-volatilized particles, and then enters a 15 $cm^3$ mixing volume prior to entering the CRDS cavity. There, $O_3$ (~30 ppmv after dilution) is added to the sampled air to convert any NO that formed in the thermal dissociation to $NO_2$. As the rate constant for the $NO + O_3 \rightarrow NO_2 + O_2$ reaction is more than three orders of magnitude faster than the $NO_2 + O_3 \rightarrow NO_3 + O_2$ reaction, conversion of $NO_2$ to $NO_3$ (and

subsequently to $N_2O_5$) is at most 1-2% in this mixing volume and is corrected for using a previously described method (Fuchs et al., 2009). To measure the thermograms shown in this paper, the oven temperature was set to a sequence of temperatures spanning 300 to 650 °C and spaced by 25 °C in a random order. The measured $NO_2$ concentrations are averaged at each temperature set point for approximately 10 – 15 minutes.

**2.2 $NO_y$ samples and additions**

Samples of reactive nitrogen species (labeled as "$NO_y$ source" in Fig. 1) were introduced into the TD oven in several ways. $HNO_3$ and $NH_3$ were obtained by passing a 50 sccm flow of zero air through a calibrated 45 °C permeation tube containing $HNO_3$ (VICI Metronics) or $NH_3$ (KinTek), providing gaseous outputs of 64 and 23 ng/min, respectively (Neuman et al., 2003). Subsequent dilution in 0.5 - 4

slpm zero (synthetic) air resulted in $HNO_3$ and $NH_3$ concentrations of 5 to 40 ppbv. Because both these species readily adsorb to instrument surfaces (Neuman et al., 1999), only FEP Teflon tubing was used between the permeation tube and the TD oven, all tubing was kept as short as possible (typically less than 30 cm) and was wrapped in 100°C heating tape to reduce losses to the walls. However, these precautions





were found to be unnecessary in this laboratory study, since the constant flow from the permeation tube resulted in an equilibrium in which the adsorption losses to the walls were equal to the rate of off-gassing.

NO was obtained by dilution of the output of a calibrated standard (Scott-Marin, 0.2% in $N_2$). $N_2O_5$ was synthesized via a procedure adapted from Davidson et al. (1978) and Bertram et al. (2009),

which has been used as a calibration for the $N_2O_5$ channel of a CRDS $NO_3$ instrument (Dubé et al., 2006;Wagner et al., 2011). Pure samples of NO and $O_2$ were mixed to yield $NO_2$, and this mixture was reacted in a flow tube with excess $O_3$, yielding $NO_3$ which then reacted with $NO_2$ to form $N_2O_5$. The resulting mixture flowed through a glass trap at -78°C, where $N_2O_5$ solidified as a white crystal. A gaseous sample of $N_2O_5$ was obtained by flowing 20 - 50 sccm of zero air over the solid -78 °C sample,

and then diluting further in zero air. Gas phase $N_2O_5$ prepared in this way is known to contain variable but significant amounts of $HNO_3$ (Bertram et al., 2009), and thus efforts were made to minimize this interference by baking all glassware for several hours before use, and by distilling the solid $N_2O_5$ sample regularly by bringing it to room temperature under an $O_3$ flow for 10 minutes. Nevertheless, some $HNO_3$ was always present in the sample, and therefore the output of the trap was passed through a nylon wool

scrubber prior to entering the TD oven, which removed $HNO_3$ without significantly perturbing the $N_2O_5$ concentration. Finally, ammonium nitrate particles were generated by running a 0.1 g/L solution of aqueous $NH_4NO_3$ through an atomizer and size-selecting particles of a certain diameter with a custom-built differential mobility analyzer (DMA). Conductive tubing, rather than Teflon, was used to minimize electrostatic build up and loss of particles to the walls before entering the TD oven.

In order to test whether common atmospheric gases would interfere with the conversion efficiency, some additional species were added to the sample prior to entering the oven. Water was added by bubbling the dilution zero air through a water bubbler prior to mixing with the $HNO_3$ sample. Various amounts of $O_3$ were added by running the dilution zero air through an $O_3$ calibrator (Thermo Scientific 49i) that is also capable of generating up to 200 ppm $O_3$ in 1 - 3 slpm of zero air. We also investigated the

effect of various VOCs, including a high concentration of propane (~5 ppmv) and a standard mixture of VOCs (Air Liquide) consisting of n-hexane (1.234 ppm), propanal (0.397 ppm), 2-butanone (1.237), benzene (1.151 ppm), methylcyclohexane (0.938 ppm), ethylbenzene (1.213 ppm), 2,2,4-trimethylpentane (1.186 ppm), isopropyl benzene (1.148 ppm) and ethanol (0.994 ppm). This mixture is commonly used to calibrate GC/MS instruments, but here provides common atmospheric species with a range of masses,

bond strengths, and degrees of oxidation. It was diluted to 50 ppbv total VOCs by addition of zero air prior to entering the oven. We also added CO in varying quantities to the $HNO_3$ and $NH_3$ samples.



### 2.3 Ancillary measurements

Several instruments were used as ancillary confirmation for some of the $NO_y$ sample concentrations. In each case, a Teflon tee split the sample input and a portion of the flow was pulled into the secondary instrument prior to entering the TD oven, as shown in Fig. 1. In the case of $NH_3$, a Picarro
G2103 $NH_3$ Analyzer with a manufacturer's specified 1 ppbv detection limit at 5 second integration time was used. A custom-built iodide adduct chemical ionization mass spectrometer (Lee et al., 2014), described in further detail in (Veres et al., 2015), was used to monitor the $N_2O_5$ and $HNO_3$ concentrations from the $N_2O_5$ solid sample prior to dissociation in the oven. In this instrument, $N_2O_5$ and $HNO_3$ mixed with $I^-$ ions produced by passing $CH_3I$ through a $^{210}Po$ source, and the resulting $HNO_3 \cdot I^-$ and $N_2O_5 \cdot I^-$ ions
were detected by quadrupole mass spectrometry at m/z = 190 and 235 respectively. Lastly, an ultra-high sensitivity aerosol spectrometer (Droplet Measurement Technologies) was used to monitor the size distribution of the size-selected ammonium nitrate particles (Cai et al., 2008).

HNO_3 and $NH_3$ conversion efficiencies were also tested using ambient air for dilution (rather than synthetic air), as sampled during daytime in August 2016 in Boulder, CO. Ambient air was drawn into the
two of the four channels of the NOAA TD-CRDS instrument, through two side-by-side identical quartz ovens heated to 650 °C at a flow rate of 1.4 slpm, and the output of either the $NH_3$ or $HNO_3$ permeation tube was inserted directly into the exposed inlet of one of the ovens, for a duration of approximately 6 minutes. The $NO_2$ signal was measured by one of the remaining channels in the NOAA TD-CRDS instrument, and the conversion efficiency of each species was calculated by comparing the difference in
$NO_2$ signal between the two ovens relative to the calibrated output of the permeation tube, to correct for small differences in $NO_2$ signal between the two ovens.

We also present results measured in the Berkeley TD-LIF instrument. It is described in greater detail elsewhere (Day et al., 2002), but briefly, $HNO_3$ and *n*-propyl-nitrate samples were provided by permeation tubes similar to those described in Sect. 2.2, diluted in dry zero air, and passed through 20 cm
heated length quartz ovens at a flow rate of 2 slpm. This resulted in residence times of approximately 50 ms. The $NO_2$ released in the thermal conversion was supersonically expanded into the detection region and measured by laser induced fluorescence from an individual ro-vibronic $NO_2$ line. The $NO_y$ conversion ratio was calculated as the measured $NO_2$ concentration relative to the maximum $NO_2$ signal at high temperatures, as the oven temperature was changed at a rate of -10 °C per minute.

### 2.4 Box modeling

A simple kinetic box model was used to support the experimental findings. Rate laws for ~60 reactions possibly involved in the dissociation and secondary chemical reactions of each $NO_y$ species (listed in the Supporting Information), were obtained from the JPL Kinetics Database (Sander et al., 2011)



and the NIST Chemical Kinetics Database (Manion et al., 2015) at temperatures spanning the 25 - 650 °C range of the experimental thermograms. For every $HNO_3$, $N_2O_5$, and $NH_3$ thermogram, a simulation was run at each temperature, assuming a starting concentration of the $NO_y$ species equal to that observed in the experiment, and lasting the duration of the residence time in the oven. The simulation was then

allowed to keep running at room temperature for an additional ~1 second to mimic the conditions between the oven and the instrument. During this additional low temperature time, 30 ppmv of $O_3$ was added to the simulation to convert NO to $NO_2$ as in the TD-CRDS instrument. The final concentration of $NO_2$ at the end of the simulation was recorded for each temperature, which resulted in a simulated thermogram. Several simplifying assumptions were made here. We assume instantaneous heating and cooling of the

sample, and a uniform temperature profile along the 38 cm length of the TD oven. We also only consider gas-phase reactions, and neglect any surface-mediated reactions. When possible, JPL-recommended values for the rate constants were used, but many of those listed did not span the full temperature range of the thermograms. When JPL values were not available, rate laws from the NIST database were used (see Table S1). We also derive temperature-dependent wall loss constants for O and OH using the procedure

outlined by Thieser et al. (2016), but find that better agreement in some simulations can be achieved with the experimental data by using an empirical value, or no wall loss at all. As can be seen in Sect. 3, these simulations successfully replicated a major portion, but not all, of the experimental results, likely due to these simplifications.

**3 Results**

**3.1 HNO3 thermograms**

Figure 2 shows the conversion efficiency of $HNO_3$ to $NO_2$ as a function of temperature, for several flow rates through the oven. Conversion efficiency was calculated as the measured $NO_2$ mixing ratio divided by the input $HNO_3$ mixing ratio. The box model simulations for each flow rate are shown as solid lines of corresponding color. The $HNO_3$ permeation tube has a calibrated output of 64 ng/min, which corresponds

to an expected $HNO_3$ concentration of between 5 and 40 ppbv, depending on the zero air dilution required for each flow rate. The output of the permeation tube was found to contain approximately 2.5% $NO_2$ and all $HNO_3$ thermograms have had this 2.5% baseline signal subtracted. At a flow rate of 1.9 slpm (where the oven residence time is 30 - 100 ms depending on temperature), we observe 100% conversion of $HNO_3$ at oven temperatures above 600 °C, whereas the thermograms obtained at 1 slpm and 3 slpm reach a

maximum conversion of 100% at 550 and 650 °C respectively. The 0.5 slpm thermogram has a slightly lower maximum conversion efficiency (95%), possibly due to recombination reactions during the extended time in the cool down region prior to detection.



The box model simulations in Fig. 2 mimic the shape of the experimental data, but some are slightly shifted to higher or lower temperatures, likely because the simulation is extremely sensitive to the flow rate and may be affected by the simplifying assumptions detailed in Sect. 2.4. The shape of the simulated thermogram is entirely controlled by the rate law of the initial dissociation reaction of $HNO_3$ to

NO₂ + OH. This reaction has a third order rate constant of $k_0(T) = 1.82 \times 10^{-4} \cdot (T/298)^{-1.98} \cdot e^{(-24004/T)}$ and a high-pressure limit of $k_\infty(T) = 2 \times 10^{15} \cdot e^{(-24054/T)}$ (Glänzer and Troe, 1974) and thus at a midrange temperature such as 500 °C, the $HNO_3$ lifetime is approximately 250 ms. The inner volume of the oven is 4.5 cm³, and so at a flow rate of 1.9 slpm, the gas has a plug flow residence time of 38 ms in the 500 °C oven, compared to a residence time of 77 ms at 1.0 slpm and 153 ms at 0.5 slpm. The simulated conversion efficiency in these mid-range temperatures is therefore extremely sensitive to the flow rate, in agreement with our experimental results. However, the experimental 100% conversion efficiency at high temperatures indicates that there is virtually no recombination of OH and NO₂ once formed, because the recombination rate for OH + NO₂ is quite low, and because OH radicals are far more likely to be lost to the walls of the oven (at a diffusion-limited rate determined by Day et al. (2002) of ~46 s⁻¹ for 1/4" OD tubing). This is in contrast to ANs and PNs, for which the reaction of the dissociated peroxy and alkyl radicals with NO₂ is a significant interference (Thieser et al., 2016), but in good agreement with the $HNO_3$ results of Day et al. (2002).

At a flow rate of 1.9 slpm, we observe a ~6% conversion of $HNO_3$ to NO₂ at an oven temperature of 400 °C. Although this efficiency is specific to the conditions of the oven used here, it is a key finding since 400 °C is in the vicinity of the temperature set point chosen for selective detection of total alkyl and multifunctional nitrates by TD-LIF (Day et al., 2002) and other TD instruments. This result is in good agreement with Thieser et al. (2016), who found a ~10% $HNO_3$ conversion at 450 °C. Sadanaga et al. (2016) report ~15% $HNO_3$ conversion at 360 °C at a TD residence time of 3.4 sec, which exceeds the range of our study but follows the trend in Fig. 3. In a previous study (Wild et al., 2014), we presented thermograms designed to demonstrate quantitative conversion efficiency at high temperatures. The temperature dependence of thermal conversion was not well constrained at lower temperatures, and showed, for example, 30% conversion at 400 °C. As discussed by Sobanski et al. (2016), the large conversion efficiency presented by Wild et al. (2014) at this temperature is likely incorrect. The extent of $HNO_3$ conversion is dependent on the residence time in the oven, but because residence time for a given flow rate changes with oven temperature, it is easier to observe this effect by plotting conversion efficiency versus residence time, as in Fig. 3, for five different temperatures (350, 400, 450, 500, and 600 °C). This plot represents transects through Fig. 2 at these five temperatures. Figure S2 shows a log scale plot to highlight the low conversion efficiency region. Most instruments utilizing the TD oven technique use a set point between 350 and 450 degrees and a residence time between 30 and 100 ms to selectively





detect ANs and not $HNO_3$ (Day et al., 2003;Paul et al., 2009;Thieser et al., 2016), but Fig. 3 demonstrates that there is significant variability in the $HNO_3$ conversion efficiency that depends nonlinearly on oven residence time.

We further measure the effect of pressure on the conversion by placing a heated stainless steel needle valve in front of the oven, thus lowering the pressure inside the oven to 250 mbar. The low pressure transects for each of the five temperatures can be seen in open circles in Fig. 3, and the full thermograms are displayed in Fig. S3. The low pressure transects are slightly lower than those at ambient pressure for the 450 and 500 °C setpoints, but match reasonably well at low and high temperatures, indicating that the onset and final conversion of $HNO_3$ are not strongly sensitive to pressure. To ensure that $HNO_3$ was not lost on the walls of the stainless steel valve, the conversion efficiency was measured with the valve fully open, and was found to match that taken with no valve. These experiments demonstrate the importance of verifying that a given temperature set point and flow rate is suitable for measurement of alkyl nitrates without interference from $HNO_3$ conversion.

To demonstrate the variability within individual TD ovens, an example of the $HNO_3$ conversion efficiency near the alkyl nitrate temperature setpoint, as measured by the Berkeley TD-LIF instrument, is shown in Fig. 4. This inlet's alkyl nitrate setpoint temperature was chosen to be just past the plateau in the n-propyl-nitrate signal at 410 ºC. The $HNO_3$ conversion to $NO_2$ was found to be 2.5%, which for most TD-LIF experiments would be negligible compared to other uncertainties in measured ANs (±15%) and no correction was applied. One example where a correction was significant was for the NASA DISCOVER-AQ California deployment, which took place in California's central valley during a period of high $NH_4NO_3$ aerosol loading. Ratios of $HNO_3$ to ANs were high enough that a correction was necessary and applied to both observations (Pusede et al., 2016). As $HNO_3$ is derived by subtraction of the ANs, any $HNO_3$ conversion at the AN temperature results in a high bias for ANs and an equal low bias for $HNO_3$. The sum of the two remains correct independent of the onset of the $HNO_3$ conversion. The Berkeley group has found the $HNO_3$ conversion to be oven dependent even for identical pressure and flow conditions indicating some but not all ovens have impurities at the walls that effectively catalyze $HNO_3$ decomposition. Ovens with high $HNO_3$ conversion efficiencies at low temperatures were discarded. These results highlight the importance of careful evaluation and calibration of each TD oven, even when the inner volumes and flow rates are similar.

**3.2 $HNO_3$ thermograms with additions**

Tests for other interferences to $HNO_3$ and AN measurements included adding several different chemical species to the $HNO_3$ sample prior to entering the oven. These were designed to test the hypothesis that certain trace gases found in ambient air would interact with radicals in the oven, or would





themselves dissociate to form radicals which could react with NO, $NO_2$, OH, or $HNO_3$. The results are shown in Fig. 5. In Fig. 5a, a portion of the dilution air was passed through a distilled water bubbler prior to diluting the $HNO_3$, bringing the relative humidity up to 66%. The change in RH does not alter the shape, onset, or total conversion efficiency of the thermogram. This is to be expected, as the oven

temperature is not high enough to dissociate $H_2O$ to OH + H, and reactions between $H_2O$ and the relevant species formed in the oven from $HNO_3$ dissociation are far too slow to be important here. However, it should be noted that both $H_2O$ and $HNO_3$ are sampled in this experiment at a steady concentration, and it is possible that during ambient sampling, rapid changes in the RH or $HNO_3$ concentration could change the overall efficiency. Figure 5b shows the measured thermogram with the addition of ~50 ppbv VOCs

(described in Sect. 2.2) with and without the addition of 90 ppbv $O_3$, as well as the addition of 5 ppmv of propane. If organic radicals were produced thermally in the TD oven, they could potentially react with $NO_2$, thus altering that signal. However, the bond dissociation energy of the C-H or C-C bonds most likely to thermally dissociate in each of the VOCs are all significantly higher (typically >100 kcal/mol) than that of the O-N bond in $HNO_3$ (~50 kcal/mol) making it unlikely that organic radicals are formed

inside the oven from dissociation of VOCs. Reactions of unsaturated hydrocarbons with O atoms or OH radicals tend to be rapid and would produce organic radicals, but these would likely only react with $NO_2$ to form ANs or PNs. The oven is set at sufficiently high temperatures to dissociate ANs and PNs back to $NO_2$ + the organic radical. Addition of these VOCs does not affect the measured conversion efficiency, even in the presence of ambient levels of $O_3$. Ozonolysis of the unsaturated hydrocarbons is slow enough

(typically on the order of 1 x $10^{-17}$ $cm^3$ $molecule^{-1}$ $s^{-1}$) to not have any effect here (we would expect < 0.0001% reaction for the duration of the oven residence time). An extremely high concentration of propane also has no effect on the overall conversion efficiency, within the error bars of the measurement, for the same reasons as detailed above.

Figure 5c shows the addition of both small and large quantities of $O_3$ to the $HNO_3$ sample. Small

quantities do not change the onset or overall conversion efficiency, but larger amounts of $O_3$ reduce the conversion efficiency at high temperatures. The kinetic box model does not predict this reduction, as it predicts 100% conversion efficiency to $NO_2$ at all $O_3$ levels. The dominant reaction of $O_3$ in the model is the reaction with $NO_2$ to make $NO_3$, but since this reaction is occurring at high temperature, any $NO_3$ formed will immediately dissociate to $NO_2$ (see Sect. 3.2). $O_3$ also thermally dissociates to O + $O_2$ at

temperatures above 200 °C (see Fig. S4), but the dominant fate of the O radicals should be loss to the walls. Of the O atoms that are not lost to the walls, their primary reaction is also with $NO_2$ to form $NO_3$. Nevertheless, there is an apparent reduction in the conversion of $HNO_3$ to $NO_2$ with increasing $O_3$. While the $O_3$ concentration range in Fig. 5c exceeds that found in ambient air, highly polluted areas may have large enough $O_3$ concentrations to make this reduction in conversion efficiency significant. Finally, the





addition of 400 ppmv CO in Fig. 5d has a marked effect on the onset, shape, and final conversion of the HNO$_3$. This addition was tested because gold catalytic NO$_y$ converters require a 1% CO addition to drive the dissociation forward. We find that ~0.5% CO is sufficient to promote HNO$_3$ dissociation even in the absence of a gold catalyst. However, our kinetic model does not replicate the results of the CO addition.

Since the rate-limiting step in these thermograms is the initial dissociation of HNO$_3$, it is unlikely that the reaction between CO and OH or NO$_2$ plays a role here. It must therefore be caused by a reaction which changes the rate kinetics of the initial dissociation step. However, to our knowledge there have been no laboratory kinetics studies on the CO + HNO$_3$ reaction. It is possible that there is some surface reaction that affects the HNO$_3$ conversion in the presence of CO.

We also note that previous work on TD ovens (Day et al., 2002;Thieser et al., 2016) has cautioned that the elevated temperature of the oven may accelerate the reaction between ambient levels of NO and O$_3$ to generate NO$_2$, thereby creating NO$_2$ signal that is in fact due to ambient levels of NO. This issue does not affect the TD-CRDS NO$_y$ detection scheme, as excess O$_3$ is intentionally added to the mixing volume after the oven to convert NO to NO$_2$ to measure total NO$_y$. Nevertheless, we have

investigated how NO responds in the oven, and the results are shown in Fig. S5. A 15 ppbv NO sample was passed through the oven. When no excess O$_3$ is added to the mixing volume, no NO$_2$ signal is seen, and when mixing volume O$_3$ is added, full conversion of NO to NO$_2$ is observed, as expected. However, when 100 ppbv of O$_3$ is added to the oven (with no mixing volume O$_3$ addition), approximately 2.2 ppbv NO$_2$ signal was observed, or a 15% conversion. This is consistent with the kinetic rate laws for NO + O$_3$

and NO + O, but we do not differentiate between these two mechanisms in these experiments, as O$_3$ will always form O at the elevated oven temperatures.

### 3.2 N$_2$O$_5$ thermograms

Figure 6 shows the measured thermogram of N$_2$O$_5$ at ambient pressure and flow rates of 1.9 and 1.0 slpm, with the kinetic model simulations for each flow rate shown in solid and dashed lines. Two distinct

dissociation steps are observed and confirmed by the kinetic model, one between 30 and 110 °C corresponding to the dissociation of N$_2$O$_5$ to NO$_2$ + NO$_3$, and one above 300 °C corresponding to the dissociation of NO$_3$. The N$_2$O$_5$ synthesis method also produces HNO$_3$ (Bertram et al., 2009) and because the bond enthalpies of NO$_3$ and HNO$_3$ dissociation are similar (both ~50 kcal/mol), the thermograms of these two species are expected to overlap at high temperatures. Thus a nylon wool scrubber was used to

remove HNO$_3$, and the scrubbed sample was simultaneously monitored with an iodide chemical ionization mass spectrometer, described in Sect. 2.3, to ensure the HNO$_3$ (and not the N$_2$O$_5$) was completely removed. The flow rate was lowered to 1.0 slpm in the high temperature scans to accommodate both instruments with better signal-to-noise. The CIMS measured approximately 120 pptv





$HNO_3$, possibly due to hydrolysis of $N_2O_5$ after the scrubber, and thus more than 99.5% of the $NO_2$ signal we observe is attributed to $N_2O_5$.

At high temperatures, each $N_2O_5$ is expected to produce two $NO_2$ molecules. Conversion efficiency is calculated from the measured $NO_2$ concentration relative to the $N_2O_5$ concentration measured

by the CIMS instrument, which samples prior to the TD oven. However, the CIMS instrument requires an empirical calibration factor for any species it measures, and while the $HNO_3$ signal may be calibrated using the permeation tube described in Sect. 3.1, there was no independent calibration available for $N_2O_5$ – only the signal measured using the TD-CRDS instrument. Therefore, the CIMS $N_2O_5$ signal was assumed to correspond to a 200% conversion efficiency in the TD-CRDS at 650 °C, and the relative

conversion was measured at lower temperatures. The first dissociation step of $N_2O_5$ to $NO_2$ and $NO_3$ is expected at oven temperatures above 110 °C, but because the sample must then travel through a "cool down" region prior to entering the CRDS optical cavity (see Fig. 1), approximately 10% of the $NO_2$ and $NO_3$ is expected to recombine back to $N_2O_5$, based on the rate constant and the residence time in the mixing volume. This behavior has been well characterized previously (Fuchs et al., 2009) and is

accounted for in the data analysis, and as expected, we observe a 91% conversion efficiency of $N_2O_5$ to $NO_2$ between 110 and 300 °C. At higher temperatures, $NO_3$ dissociates in the oven before recombining with $NO_2$, and thus a 200% conversion efficiency is observed. While this is not an absolute measure of conversion efficiency, the relative conversion efficiency is consistent with $N_2O_5$ dissociation and recombination reaction rates to generate two $NO_2$ molecules in a distinct stepwise manner. At 150 °C and

400 °C, the temperature setpoints often used for detection of PANs and ANs, we find 90% and 105% conversion of $N_2O_5$ to $NO_2$, respectively. The exact values are highly dependent on the residence time in both the oven and in the cool down region, but serve to highlight the importance of characterizing the $N_2O_5$ response in every thermal dissociation oven.

We also measured the conversion of $N_2O_5$ without the mixing volume $O_3$ addition at two relevant

temperatures, in order to determine the mechanism for $NO_3$ dissociation. These data are shown in green triangles in Fig. 6, and show no difference in onset or maximum conversion efficiency whether or not mixing volume $O_3$ is added. As the mixing volume $O_3$ converts ambient or thermally produced NO to $NO_2$, the similarity of the two spectra indicates that the $NO_3$ dissociation mechanism must be $NO_3 \rightarrow NO_2$ + O. However, there are no published rate laws for this reaction and the few studies on $NO_3$ thermal

dissociation have disagreed about whether the reaction proceeds to $NO + O_2$ (Johnston et al., 1986) or $NO_2$ + O (Schott and Davidson, 1958). The former argued for the $NO_3 \rightarrow NO + O_2$ mechanism based on thermodynamics, as this reaction is exothermic. However, this implies that $NO_3$ would be thermally unstable at room temperature, which is not the case. It is likely that there is a significant energy barrier to this reaction. The bond enthalpy of the $NO_3 \rightarrow NO_2$ + O reaction, on the other hand, is 50.4 kcal/mol,


nearly identical to that of HNO$_3$ → NO$_2$ + OH, and the two thermograms are very similar in shape and are centered at the same temperature (500 °C). The simulation shown in Fig. 6 is a fit rate law of k(T) = 1 x 10$^{-2}$·(T/298)$^9$·exp(-1500/T) obtained by taking the rate law of HNO$_3$ dissociation and iteratively adjusting it until it matched the data. Essentially identical results were observed in the TD-LIF instrument. (Cohen,

5      2016).

### 3.3 NH$_4$NO$_3$ thermograms

NH$_4$NO$_3$ particles were generated in situ from an aqueous solution, dried, and size-selected by a differential mobility analyzer (DMA) set at 250 nm prior to entering the TD oven. The conversion efficiency was calculated by comparing the measured NO$_2$ concentration in the TD-CRDS instrument to

the expected number of NH$_4$NO$_3$ molecules in the aerosol particles, derived from the number and size of the aerosol particles as measured with an ultra-high sensitivity aerosol spectrometer (UHSAS). The measured UHSAS histogram was used, along with the literature value for the density of NH$_4$NO$_3$, to convert particle diameter to particle volume, and then to the total number of NH$_4$NO$_3$ molecules. We demonstrate here that the dissociation pathway is NH$_4$NO$_3$ → NH$_3$ + HNO$_3$, and we assume that NH$_3$ is

not converted in any significant fraction. A temperature-dependent baseline NO$_2$ signal is observed when the DMA voltage is set to zero (i.e. when no particles are transmitted) which is attributed to gas-phase HNO$_3$ molecules which have evaporated from the particles and adsorbed to the tubing walls, and which are subtracted from the total signal. Figure 7 shows the measured thermogram of NH$_4$NO$_3$ with the thermogram of gas phase HNO$_3$ from Fig. 2 overlaid. The close agreement between the two thermograms

demonstrates that the dissociation pathway is NH$_4$NO$_2$ → NH$_3$ + HNO$_3$.

For particles that pass through the DMA at a given size setpoint, the UHSAS measures a size histogram that peaks at a diameter approximately 8% lower, likely because the NH$_4$NO$_3$ particles are slightly non-spherical, and therefore the electrical mobility diameter is slightly larger than the geometric diameter. This phenomenon has been discussed at length elsewhere (DeCarlo et al., 2004), and we make

no attempt to further characterize NH$_4$NO$_3$ particle behavior in the DMA – we have simply taken the UHSAS histogram data to calculate the particle volume, even though this is also subject to slight differences based on the refractive index of NH$_4$NO$_3$. However, if the TD oven failed to volatilize and convert all NH$_4$NO$_3$ particles to HNO$_3$ and then to NO$_2$, the measured thermogram would deviate from the HNO$_3$ spectrum at lower temperatures, where perhaps the heat is not sufficient to drive the NH$_4$NO$_3$

out of the condensed phase. The close match between the two is a good indication that the conversion goes to completion. Additionally, Figs. S6 and S7 show a sample NO$_2$ measurement measured by TD-CRDS at 650 °C as the particle diameter setpoint is changed. There is no correlation between particle size



and conversion efficiency, indicating that the oven is completely converting all particles without a size dependence.

### 3.4 $NH_3$ thermograms

A previous study (Wild et al., 2014), investigated whether ambient levels of ammonia would represent an
interference to $NO_y$ conversion, and found that it made at most a 1% difference to the $NO_2$ signal in dry air, but that this effect was suppressed when RH > 10%. We find in the present study that there is a significant interference when ambient levels of both $NH_3$ and $O_3$ are present in the oven, but that this effect is potentially suppressed by other species found in ambient sampling. Figure 8 shows a thermogram of $NH_3$ with and without 100 ppbv $O_3$ present in the oven. The conversion of $NH_3$ to $NO_2$ at 650 °C,
calculated as the observed $NO_2$ signal relative to the added $NH_3$ concentration, is small without $O_3$. This is consistent with the previous study of Wild et al. (2014). However, when 100 ppbv of $O_3$ is added, the thermogram reaches a maximum molar conversion efficiency of 8%, with an onset near 400 °C (red circles). In contrast to the $HNO_3$ thermograms, however, this signal does not appear to plateau at 650 °C but rather continues to grow at higher temperatures. This result is similar to the interference reported by
Dillon et al. (2002), which was attributed to a reaction between $NH_3$ and $O_3$. The interference is only present when $O_3$ is added to the mixing volume, indicating that the conversion of $NH_3$ must be producing NO, rather than $NO_2$, and is subsequently unimportant to instruments that measure $NO_2$ only, such as the Berkeley TD-LIF instrument. A kinetic model simulation of both experiments is shown in solid line in Fig. 8. This simulation was carried out with 35 relevant reactions between $NH_3$, $O_3$, and the radicals that
are formed from these two species in the oven, with the most important reactions are listed below. The reaction between $NH_3$ and $O_3$ is far too slow to be relevant here, and the oven temperature is not high enough to dissociate $NH_3$ to $NH_2$ + H ($\Delta H$ = 108 kcal/mol). However, $O_3$ dissociates readily at oven temperatures above 200 °C, and once formed, the O atoms may react with $NH_3$ to form $NH_2$.

$$O_3 \rightarrow O_2 + O \tag{3}$$

$$NH_3 + O \rightarrow NH_2 + OH$$

$$NH_3 + OH \rightarrow NH_2 + H_2O$$

The reactions of $NH_3$ are the slowest steps, but once formed, $NH_2$ reacts readily with O atoms.

$$NH_2 + O \rightarrow HNO + H \tag{4}$$

$$\rightarrow OH + NH$$

$$\rightarrow NO + H_2$$

HNO then reacts with O, OH, and H to form NO, or can also directly dissociate to form H + NO.

$$HNO + O \rightarrow OH + NO \tag{5}$$

$$HNO + OH \rightarrow H_2O + NO$$





$$HNO + H \rightarrow H_2 + NO$$

$$HNO \rightarrow H + NO$$

The OH and H atoms formed in Eq. (4) and (5) then drive Eq. (3) further. This mechanism takes place entirely in the gas phase, and does not take into account any surface-mediated reactions. Many of these

reactions have only limited published studies, so the simulation used rate constants that have not been extensively tested. Additionally, to achieve a significant conversion of $NH_3$ to NO, it was necessary to decrease the O and OH wall loss constants in the model. However, even this rudimentary simulation predicts the general shape of the experimental data, although it has a maximum conversion efficiency of just under 2%, which is below that observed in the experiment. In Fig. 9, we adjusted the amount of

added $O_3$, while monitoring the conversion efficiency of $NH_3$ to $NO_2$ at an inlet temperature of 650 °C. We find that increasing the $O_3$ increases the conversion, which is consistent with $NH_3$ + O being the limiting reaction to make $NH_2$. Figure 9 also demonstrates that the conversion of $NH_3$ is partially quenched by the addition of ambient levels (~100 ppbv) of CO, likely because the $CO + O \rightarrow CO_2$ reaction competes with those in Eq. (3). Figure 10 shows that the average conversion efficiency of $NH_3$

when measured in ambient air in Boulder, CO in August 2016 (which contains 40-60 ppbv $O_3$, >80 ppbv CO, ~15% RH, and other species) is 0.5 ± 2.4%, or zero to within the 1σ error from repeated measurements. This is in contrast to the conversion efficiency of $HNO_3$ in ambient air, shown in the upper right frame of Fig. 10, which is largely unchanged from that measured in zero air. Thus, constituents present in ambient air, such as methane, CO, and water, are possibly suppressing the conversion of $NH_3$

to NO, likely through the reaction with O atoms.

## 4 Discussion

Using a thermal-dissociation cavity ringdown spectrometer (TD-CRDS), we have quantitatively added reactive nitrogen species to the TD inlet, in order to test the efficiency of the thermal conversion of each species to $NO_2$, and the effect of any interferences from other trace gases which may be present in the

ambient troposphere. We have determined that the TD-CRDS converts $HNO_3$, $N_2O_5$, and $NH_4NO_3$ particles to $NO_2$ with 100% efficiency at temperatures above 600 °C, but that the onsets of the dissociation are highly dependent on oven residence time. Despite their similar residence times, the NOAA TD-CRDS and Berkeley TD-LIF instruments measure $HNO_3$ conversion efficiencies ranging from 2.5% to ~8% at 410 °C. It is therefore important that the oven residence time is well characterized in

instruments designed to selectively detect ANs without interference from $HNO_3$. Even two TD ovens with identical inner volumes may exhibit different response functions if they have different ratios of surface area to volume.



We find that high levels of ambient $O_3$ (>500 ppbv) and CO (>400 ppmv) significantly changed the final conversion efficiency and the onset of the conversion, respectively, of the $HNO_3$ thermogram, but that ambient levels of a group of representative VOCs and high RH did not affect the measured thermogram. Modest levels of $O_3$ converted a portion of $NH_3$ to $NO_2$. The conversion mechanism likely

arises from a gas-phase reaction between oxygen atoms and $NH_3$ which produces NO. To our knowledge, the $NH_3 + O_3$ reaction in TD ovens has not been studied in detail, but previous studies of $NH_3$ conversion in catalytic converters have noted similar results to those presented here – water and CO suppress the $NH_3$ conversion to NO, while $O_3$ enhances it (Fahey et al., 1985;Kliner et al., 1997). If not quenched by other species present in ambient air, this effect could represent a potentially significant interference in field

sampling for instruments that are sensitive to NO directly, or via conversion to $NO_2$. For example, at 50 ppbv $O_3$, the 6% conversion of $NH_3$ would present an interference of more than 10% if $NH_3/NO_y$ > 1.7, which is not an uncommon condition in agricultural regions. This signal was suppressed in ambient air, indicating that $NH_3$ may not interfere with $NO_y$ under most conditions. However, ambient air in Boulder is not representative of all sampling conditions, and since the species responsible for quenching the

reaction remains unclear, more work must be done to better understand the mechanism of the $NH_3/O_3$ thermal reaction. This result, along with the others detailed above, serve to emphasize that great care must be taken to characterize the potential interferences in TD $NO_y$-conversion ovens.

The measured $N_2O_5$ thermogram exhibits a double dissociation curve, corresponding to the initial dissociation of $N_2O_5$ to $NO_2$ and $NO_3$, and the subsequent dissociation of $NO_3$. Our results indicate that

the mechanism of the second step is $NO_3 \rightarrow NO_2 + O$, in contrast to earlier literature that reported $NO_3 \rightarrow NO + O_2$ as the dominant mechanism. To our knowledge, this is the first published thermogram of $NO_3$. $N_2O_5$ is not typically considered in the TD-$NO_2$ instrument literature because the existing instruments have largely operated in the daytime, when concentrations of $N_2O_5$ are not a significant fraction of $NO_y$. The first dissociation is approximately quantitative at the oven set temperatures used to quantify PN, with

the second dissociation occurring at temperatures used for $HNO_3$ detection. However, this interference would only be significant during nighttime or during very cold weather sampling and is already accounted for in the analysis of these instruments. (Slusher et al., 2004;Thieser et al., 2016)

The thermogram of particulate ammonium nitrate matches the thermogram of $HNO_3$, within the margin of error of the UHSAS measurement. TD ovens have not typically been used explicitly for particle

detection, with a few exceptions (Voisin et al., 2003;Smith et al., 2004;Rollins et al., 2010). Fine particles likely will be sampled by the inlet, unless they are excluded aerodynamically or physically. These results demonstrate that the volatile portion of the particulate nitrates will be driven into the gas phase at low oven temperatures, consistent with Rollins et al. (2010), who used a denuder to remove gas phase nitrates and to detect aerosol organic nitrates in a 325 °C oven. Other $NO_3$ salts might also be detected via thermal





dissociation, although it is expected that they would be non-volatile at the temperatures of these TD-inlets. Bertram and Cohen (2003) examined $NaNO_3$ and determined that those particles would not be detected in TD inlets. However, these studies measured pure aerosols, and results may vary with heterogeneously mixed particles with multiple components. The initial dissociation of $NH_4NO_3$ will

produce an $NH_3$ molecule in addition to an $HNO_3$ molecule, which means that particles may be subject to the same $NH_3/O_3$ interference when sampling in ambient air, which was not considered in this study. Additionally, the particles sampled in this paper were generated and injected directly into the inlet. The efficiency of particle sampling in ambient air will depend on particle size and inlet design, particularly during aircraft measurements.

Based on the results of this paper, we make the following three recommendations: *(1)* TD ovens should be calibrated with the appropriate reactive nitrogen compounds regularly at the oven set points using the oven residence time and gas pressure that will be used in ambient sampling. *(2)* In addition to the AN and PN calibrations recommended by (Day et al., 2002;Thieser et al., 2016) and others, these calibrations should include $HNO_3$. $HNO_3$ calibration will be especially important if sampling in regions

where $HNO_3$ is in large excess over other $NO_y$ species. *(3)* Potential non-$NO_y$ species such as $NH_3$ should also be regularly introduced into the inlet under conditions where $O_3$ is present in ambient air to check for potential conversion. These recommendations are similar to those detailed in Bradshaw et al. (1998). The results of Fig. 9 indicate that calibration results may also vary significantly when sampling in ambient air, due to the large number of possible gas-phase reactions available to the wide variety of trace atmospheric

species. The last step is particularly important in instruments that detect NO as well as $NO_2$. Comprehensive calibration of these instruments $NO_y$ measurement accuracy, which in turn will provide valuable information about tropospheric $NO_x$ chemistry.

**Acknowledgements**

We wish to thank Jessica Gilman for providing the GC/MS VOC mixture, Sascha Albrecht for measuring

the $NO_2$ baseline in the $HNO_3$ permeation tube, and Jim Burkholder for the loan of a propane tank. We would also like to thank Jim Roberts, Tom Ryerson, Dave Parrish, and Joel Thornton for the helpful discussions. CCW acknowledges support from the National Research Council Research Associateship Program. The authors acknowledge support from the Atmospheric Chemistry, Carbon Cycle and Climate Program (AC4).



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

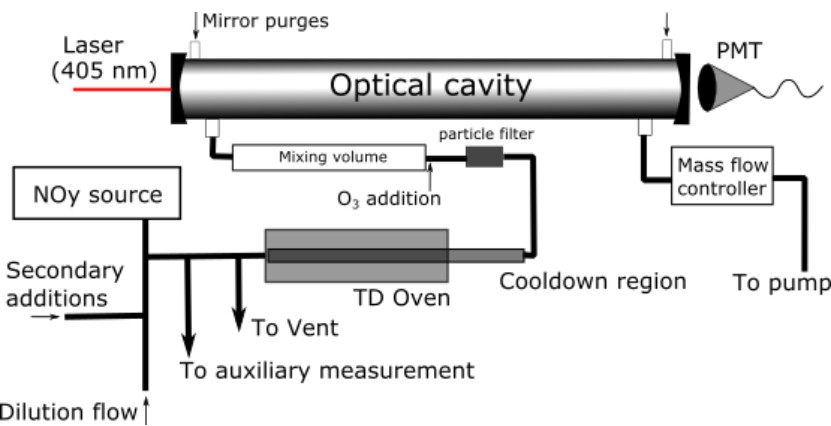

10  **Figure 1: Instrument schematic of the TD-CRDS instrument used in this study. An NOᵧ source (HNO₃ permeation tube, N₂O₅ cold trap, NH₄NO₃ particle atomizer + DMA size-selector, or NH₃ permeation tube) is diluted by a zero air flow (with an option for adding O₃, VOCs, RH, or CO through the secondary addition port), and passed through the TD oven. A portion of the flow is sampled prior to entering the oven with one of several type of auxiliary measurement (I- CIMS for N₂O₅, UHSAS for NH₄NO₃ particles, or commercial CRDS for NH₃). After flowing through a cool down region, the**
15  **sample passes through a particle filter and then is mixed with a ~30 ppmv addition of O₃ in a mixing volume before entering through the optical cavity, where NO₂ is measured by CRDS.**

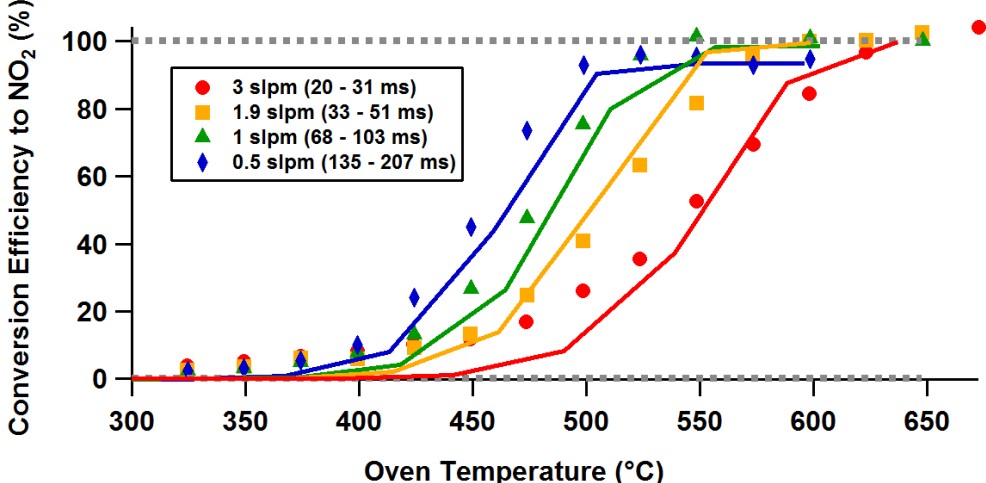

**Figure 2. HNO₃ thermograms at several flow rates. Conversion efficiency is calculated as measured NO₂ signal relative to**



the expected concentration of HNO₃. Parentheses in the legend indicate the range of residence times experienced by the sample in the heated inlet. The grey dashed lines indicate 0 and 100% conversion. Solid lines show simulations using a simple kinetic box model, as described in the text.

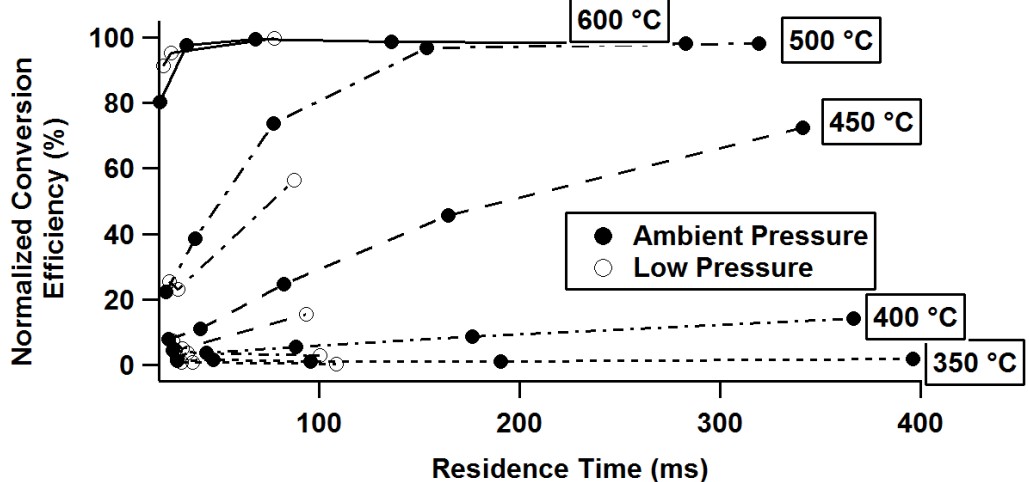

Figure 3. Conversion efficiencies of HNO₃ to NO₂ plotted as a function of plug flow residence times in the oven (see text) for 5 different temperatures. Values were obtained by scaling the measured conversion efficiency in Fig. 1 to the overall maximum and minimum of the thermogram, to account for slight differences between thermograms. Solid circles indicated measurements at ambient pressure, whereas open circles indicate measurements at low pressure. Different line traces indicated different temperatures. A temperature setpoint between 350 (small dashed line) and 450 degrees (long dashed line), and a residence time less than 200 ms are the conditions normally selected for selective detection of alkyl nitrates with no detection of HNO₃. However, under these conditions HNO₃ conversion may be anywhere between 1 and 30%.





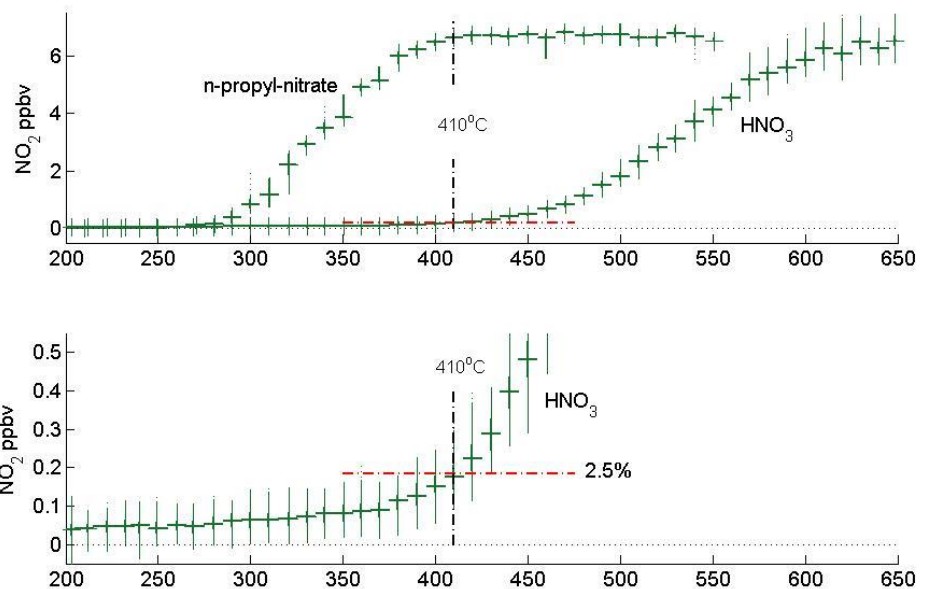

**Figure 4. HNO₃ and *n*-propyl-nitrate thermograms taken with the Berkeley TD-LIF instrument used in the NASA DISCOVER-AQ California mission. The lower panel shows only HNO₃ with the y-axis expanded to illustrate the dissociation onset. The oven is from the instrument's alkyl nitrates channel. The flow rate was approximately 2 slpm and the measurement setpoint was 410 ºC. The dataset was corrected for the 2.5% dissociation of HNO₃ in the alkyl nitrates channel. A different physical oven was used for HNO₃ at a setpoint of 620 ºC.**





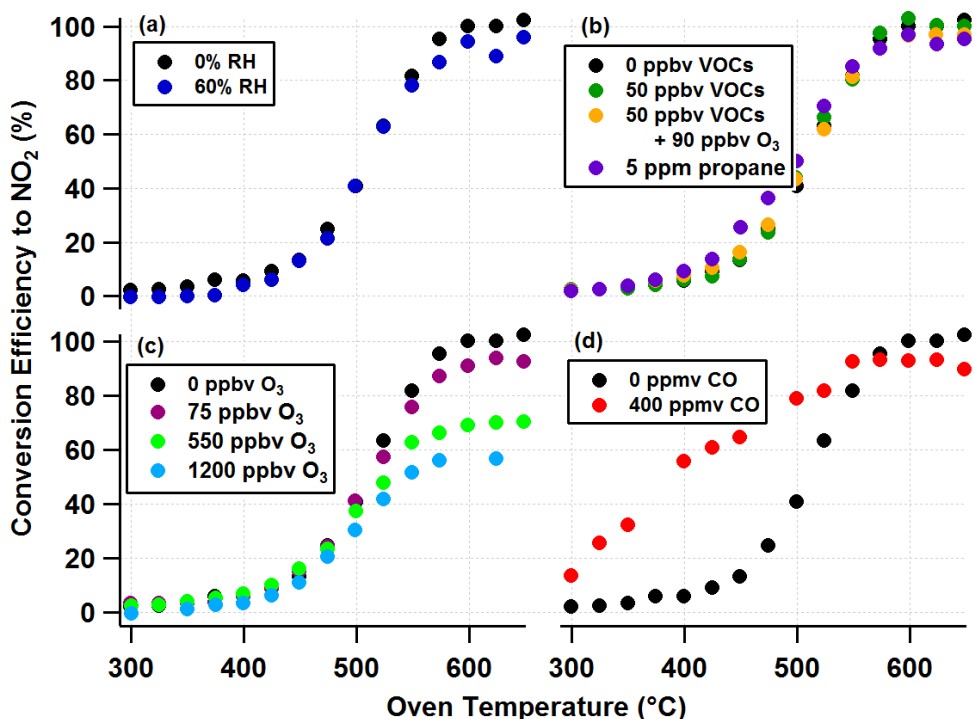

**Figure 5. HNO₃ thermograms (1.9 slpm, ambient pressure) with various additions added prior to the TD oven. In each frame, the black solid circles indicate the no-addition case. In frames (a) and (b): No effect is observed when thermogram is taken at high relative humidity or when VOCs are added. In frame (c), varying amounts of O₃ were added, ranging from ambient levels (75 ppbv) to extremely polluted levels (1200 ppbv), which decreases the overall conversion at high temperatures. In frame (d), the addition of 400 ppmv CO alters the shape of the thermogram.**





Figure 6. Thermogram of N₂O₅ at two flow rates. The red squares and red dashed line show the 1.9 slpm thermogram and simulation, while the blue circles and blue solid line show the analogous result at 1.0 slpm. The first dissociation corresponds to N₂O₅ → NO₂ + NO₃, and the second to NO₃ → NO₂ + O. The second curve reaches a maximum of 200%, while the first reaches 90 – 95%, depending on the flow rate, due to recombination of NO₂ and NO₃ in the cooldown region prior to the detector region. The black dashed line is the experimental HNO₃ thermogram from Fig. 2, offset by 100%. The green triangles indicate measurements of the conversion efficiency without the O₃ addition, confirming that the second dissociation must occur via NO₃ → NO₂ + O rather than NO₃ → NO + O₂.

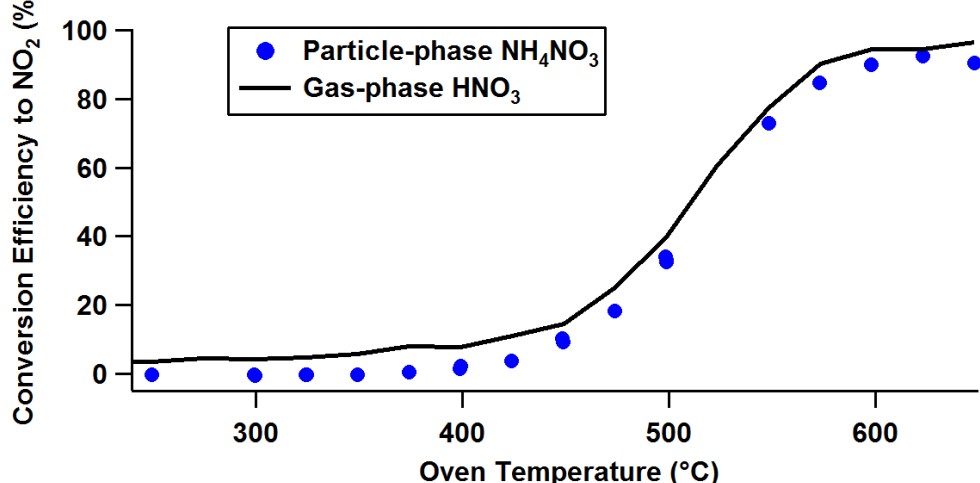

Figure 7. Measured thermogram of NH₄NO₃ particles in solid circles. The black solid line indicates the measured thermogram of gas-phase HNO₃ from Fig. 2. The close match of these two thermograms indicates that the NH₄NO₃ particles go through HNO₃ as an intermediate, and is a good indication that complete conversion is achieved.

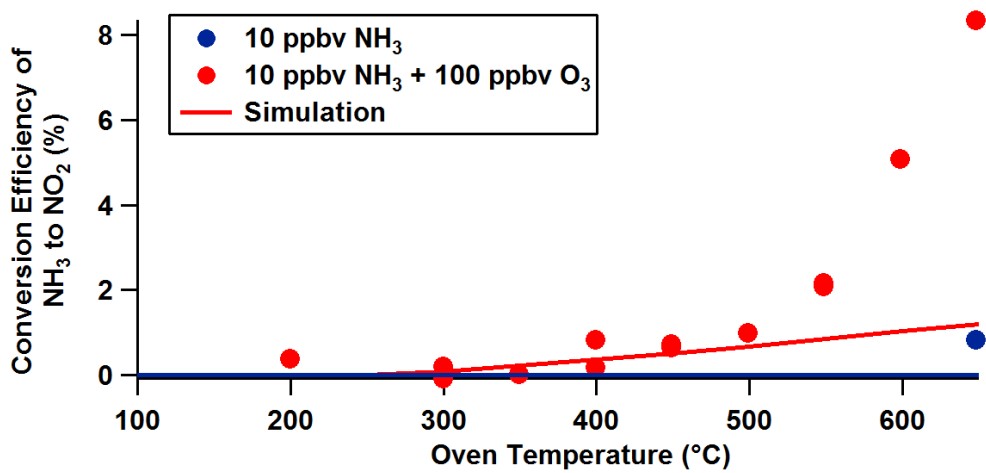





**Figure 8. Thermogram of NH₃ with 100 ppbv of O₃ added before the oven shown in red circles. The blue circle represents an analogous measurement at 650 °C with no O₃ added. Kinetic box model simulations shown in solid lines of corresponding color.**

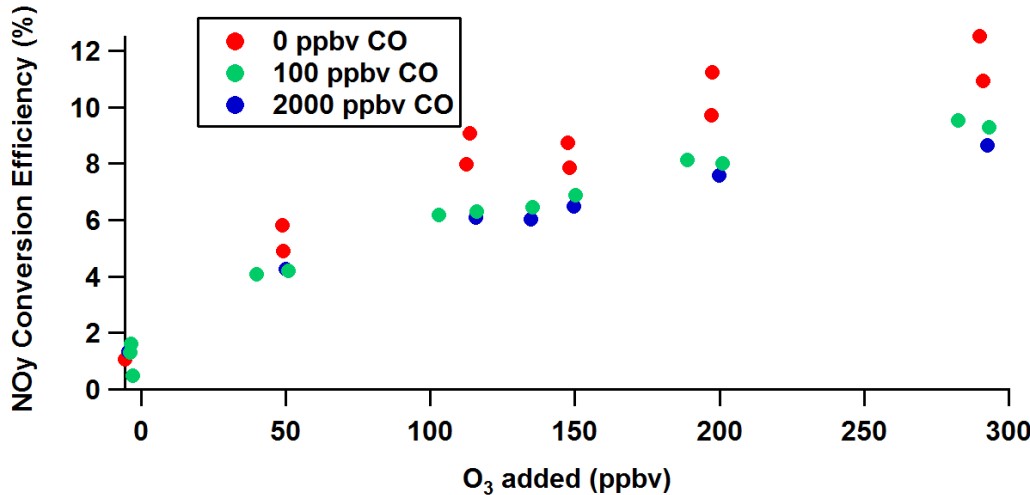

**Figure 9. Conversion efficiency of NH₃ to NO₂ as a function of O₃ added to the TD inlet. Red circles show 10 ppbv NH₃ with O₃ ranging from 0 – 300 ppbv, and the green and blue traces show similar data, but with 100 and 2000 ppbv CO added. The partial depletion of the signal (~25%) with the addition of CO indicates that the oxygen atoms formed from O₃ pyrolysis are preferentially reacting with CO instead of NH₃.**



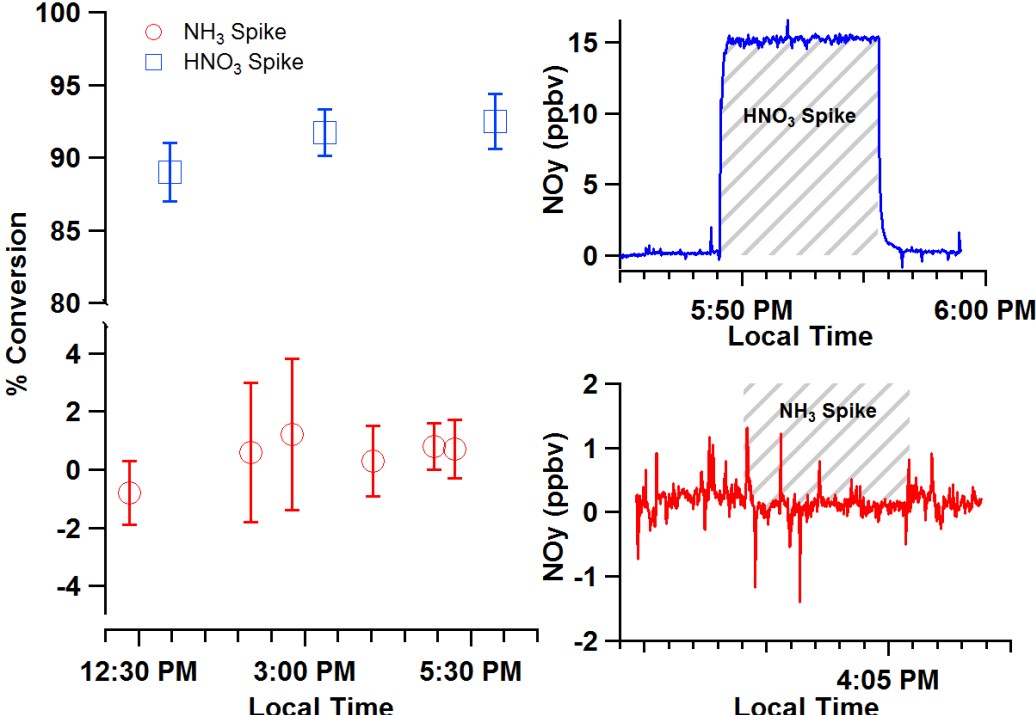

**Figure 10.** Measurement of HNO₃ and NH₃ conversion in ambient air at an inlet set temperature of 650 °C. The left panel shows measured conversion efficiencies for standard additions of HNO₃ and NH₃ to an inlet sampling ambient air in Boulder, CO on August 9, 2016. The right panels show time series of measured NO_y during standard additions. The data is the difference between two NO_y measurement channels, one with and one without the standard addition, to cancel the variation in ambient NO_y during the tests.