# Peer review of "Evaluation of the accuracy of thermal dissociation CRDS and LIF techniques for atmospheric measurement of reactive nitrogen species"

_Atmospheric Measurement Techniques, 2016_

## Referee Comment (RC1) · Anonymous Referee #1 · 19 Jan 2017

Womack et al. present an interesting set of experimental data on the efficiency of conversion of various NOy species into NO2 in TD-inlets as used in CRD and LIF based detectors of e.g. organic nitrates or HNO3. The authors have backed up the experimental data with a chemical model which is reasonably successful in reproducing the thermograms they measure. They examined HNO3, NH3 and ammonium nitrate conversion with added trace constituents such as H2O or CO or hydrocarbons. What they did NOT do was to examine the efficiency of conversion of mixtures of NOy such as HNO3 in the presence of organic nitrates, NO and NO2 as will always be the case in ambient air. I appreciate that the experiments are already complex and adding further

[Figure]

NOx containing trace gases components is not trivial. However, the extrapolation of these results to sampling real air needs to be considered. If the authors choose not to do further experiments, it would be interesting to at least predict what the effect of various levels of NO2 would be on the shape of e.g. the HNO3 thermogram (if any). For example, would addition of NO or NO2 change the NH3 / O3 thermogram (both NO and NO2 might react with the NH2 radical etc etc).

The authors may consider the following specific comments and questions when revising their manuscript.

P2L25 Methods that detect some individual components of NOy are listed. How were they selected ? Why not include e.g. NO3 or HONO etc. …..

P3L24 "For example, TD-LIF detects NO2 at low pressure following thermal dissociation, which minimizes secondary recombination reactions of the dissociated radicals". Are TD-LIF instruments always operated with the oven at low pressure which would minimise the recombination by reduction of reaction time and rate coefficient ?

P3L29 "TD-CRDS is an absolute measurement…..." Does TD-CRDS being based on a cross-section of NO2 really make it absolute ? As stated later in the manuscript, the effective optical path-length needs to be calibrated by adding known amounts of NO2. Also, the TD-inlet is part of the instrument and its dissociation efficiency needs to be calibrated (the subject of this paper).

P4L3 "though this reaction rate depends on the TD inlet pressure and flow rate". What reaction rate is this referring to (NO + O3 makes NO2 but O + NO2 or NO does not) ?

P4L10 "Thieser et al. (2016) parameterized the bias in peroxyacetyl nitrate and 2-propyl nitrate detection in their inlet as a function of ambient NO and NO2 concentrations, but noted that these parameterizations may vary for other PNs or ANs. These effects are generally considered minor compared to other uncertainties in the measurement". Is this true ? In some TD-instruments, depending on operating temperature, the

effects of radical recombination (RO2 + NO2) or oxidation of NO (RO2 + NO) can bias the detection of peroxy nitrates by factors of 2 or more and is likely the biggest source of uncertainty.

P6L11 Does the temperature probe measure the gas-temperature or the temperature of the inner wall of the quartz tube ? Can there be different due to gradients from the centre of the tube to the wall ?

P6L17 Does addition of 30 ppmv O3 have any adverse effects ? Can e.g. ozonolysis of biogenics take place in this volume ? Might this form particles or radicals (Criegees) than can react with NO2 ?

P8L7 "A custom-built iodide adduct chemical ionization mass spectrometer (Lee et al., 2014), described in further detail in (Veres et al., 2015), was used to monitor the N2O5 and HNO3 concentrations" How was the CIMS calibrated ? How accurate are the concentration measurements ?

P8L25 What is the pressure in the Berkeley oven? What are the standard operating conditions for this instrument. I believe it has been operated using different pressure configurations.

P8L16 The modelling of this system is not trivial. As the authors state, many rate constants have not been measured at the higher temperatures. Secondly, the authors do not consider surface catalysed thermal decomposition, which is important as the authors mention briefly later when discussing the low temperature NOy instruments with catalytic conversion at the metal surfaces. The wall losses of radicals is probably the biggest uncertainty and can only be assessed by variation of experimental parameters. Thieser et al. 2016 showed that variation of the concentration of the organic nitrates they were using (and thus variation of the RO2 concentration) affected the loss rate, which could then be explained using a Langmuir-Hinshelwood type expression. Did you change the concentration (e.g. of HNO3) significantly to see if the same wall loss rate constant was appropriate ? Do you expect the rate constant for wall loss to be

independent of temperature (effects of diffusion, turbulent mixing) ?

P9L23 was the HNO3 input mixing ratio based on the permeation source or the CIMS signal ?

P9L31 ..."possible due to recombination reactions......." Which ones ? Be specific.

P10L8 High and low pressure limits have been used to calculate the thermal dissociation rate constant for HNO3. What value for Fc was used to calculate the ate coefficient at 500 °C. Also, Glänzer and Troe did their study in Argon. Are the results applicable for air (what is the relative collisional stabilisation efficiency) ?

P10 L13 "...the recombination rate for OH + NO2 is quite low..." Be quantitative. What is the rate coefficient at this temperature and wat is the pseudo-first order rate coefficient for recombination for a given NO2 level of e.g. 10 ppbv. This can then be compared to the wall loss rate coefficient.

P10L27 Did Sobanski et al. (2016) also present a decomposition efficiency for HNO3 ? Are the results comparable? Note that Sobanski et al used a radical scavenger with a large surface to reduce radical recombination in the heated inlet.

P11L9 ".. the onset and final conversion of HNO3 are not strongly sensitive to pressure". Is this because wall losses are invariant with pressure ? is the conclusion that wall losses are so large that recombination never compete ? What if the sample contains not only HNO3 but also NO2 to increase the rate of re-formation of HNO3 ? The authors should consider doing one such experiment to see if elevated NO2 will influence the shape of the HNO3 thermogram. The same applies to the NH3 expeirments.

P11L26 This sentence implies that the modelling done in this study (which considers gas-phase processes only) is only a partial representation of the chemistry going on. As mentioned above, the sensitivity HNO3 detection while adding NO2 would have been useful to confirm that the simple model reproduces the thermograms for the right reasons.

P12L11 The data shows that the VOCs added had no effect. Not surprising considering their bond-dissociation energies. It would have been more informative to have added VOCs that will decompose, especially organic nitrates as they result in more complex radical chemistry and NOx.

P12L17 ". . . . .The oven is set at sufficiently high temperatures to dissociate ANs and PNs back to NO2 + the organic radical" Not true. At higher temperatures the RO2 formed from thermal dissociation of PNs is unstable (see Thieser et al. 2016).

P12L27 ". . ..The dominant reaction of O3 in the model is the reaction with NO2 to make NO3. . .." Is this true? I would have thought the pyrolysis will dominate at high temperatures. Is the O3 pyrolysis rate constant in the model correct ? What is the O-to-O3 ratio at thermal and kinetic equilibrium?

P12L30 The reaction between O and NO2 does not form much NO3 but mainly NO + O2. This is especially true at high temperatures.

P13L5 That a model with no surface-catalysed reactions cannot reproduce the effect of a surface catalysed reaction is not surprising. Why do NOy instruments with e.g. gold-surfaces see decomposition at much lower temperature than needed to break the HO-NO2 bond and why do they add CO? It is more than "possible" that surface reactions play a role, it is rather clear.

P16L25 Are there any other reactions of NH2 that should be considered. Could it react with NO or NO2 ?

P17L6. "However, even this rudimentary simulation predicts the general shape of the experimental data. . ." What aspect of the "general shape" does it reproduce ?. Perhaps you can be more concise here.

P18L3 "ambient levels of a group of representative VOCs". As already mentioned, addition of VOCs that are unstable at the inlet temperatures (organic nitrates) would have been more informative.

P18L23 "... N2O5 is not typically considered in the TD-NO2 instrument literature because the existing instruments have largely operated in the daytime..." Perhaps this statement is too general. Some instruments measure day and night and have considered effects of N2O5 thermal decomposition (e.g. Thieser et al. 2016)

P18L32 "...These results demonstrate that the volatile portion of the particulate nitrates will be driven into the gas phase at low oven temperatures.." Particulate nitrate is not only ammonium nitrate but has a large component of organic nitrates. At which efficiency will these be detected?

Technical / typographical

P2L20 Techniques that detect the major individual components of NOy include detection....

P4L1/3 negative / positive artefact = negative / positive bias ?

P5L4 Inappropriate reference. Fuchs et al were not the first to use CRDS for atmospheric trace gases as this implies.

P5L18 "light decays" ?

P5L23 "known NO2 concentrations". How were they determined? Was an "absolute method" used to measure the NO2 concentration ?

P7L22 "bubbling the dilution zero air through a water bubbler...."

P8L31 and at several other places in the manuscript. "rate laws" is the wring term. You refer to "rate constants" or "rate expressions".

References: Several have capitalsed manuscript titles. Nikitas et al spelling of Detector.

Figure captions: Figure 1: "Instrument schematic of the TD-CRDS instrument" "cool down" (sometimes cooldown" maybe you can find a better expression than "cool down region".

Figure 3: small dashed line = short dashed line ? Figure 4 "physical oven" ?? Figure 7 "in solid circles" = "as solid circles" ? Figure 8 "shown as red circles" Figure 9 delete preferentially.

Supplementary Info: Caption to Fig. S4. "...but if allowed to recombine, only ∼40% will be allowed to recombine, but that nearly all o atoms ........." Not clear what is meant here. Rewrite.

Table S1. Many/most of the reactions listed contribute little to the thermograms (e.g. does neglecting H + NO3 make any difference at all)? Please highlight those reactions that do have an influence (i.e. those that account for 90 % of the reactive flux). This would make the results of the modelling excercise more transparent. Please add (in a footnote) the original references used for the rate expressions. Just listing the NIST type label (e.g. 1986TSA) is not sufficient. What does JPL ** mean (HNO3 + OH reaction). Please mark those reactions for which experimental data in the range up to 700 celcius was NOT available and inducate which (if any) are estimated or theoretical.

[Figure]

---

## Referee Comment (RC2) · Anonymous Referee #2 · 19 Jan 2017

This manuscript presents a set of experimental data investigating the thermal dissociation of NOy species into NO2 (e.g. HNO3), with subsequent detection by CRDS and LIF. They report detailed experiments (at different temperatures) on the dissociation of HNO3, NH3, N2O5 and ammonium nitrate, all of which are either NOy species of atmospheric relevance (in the case of HNO3), or potential interferences in a total NOy instrument (in the case of NH3). The experimental work is backed up by a chemical model which does a good job at reproduced the observed thermal decomposition. The work is well presented, reasonably thorough and will of no doubt be of great interest to anyone wishing to make a total NOy measurement using the thermal decomposition

method. I would recommend publication in AMT subject to the authors answering the following comments and questions.

Could the authors explain why they did not carry out experiments using common atmospheric NOy constituents such as peroxy and alkyl nitrates? Clearly these are likely to fully dissociate at the highest temperatures and this has been shown in previous work on the TD-LIF instrument, however it would have been good to actually see this shown experimentally. A thermogram of n-propyl nitrate from the UC Berkeley TD-LIF instrument is shown but it would have been nice to have seen this for the TD-CRDS system as well (especially considering one of the conclusions from the work is that all ovens behave differently). I do not know if the authors intend to use their system to provide NOy class speciation in their atmospheric measurements (and hence need to know the thermograms for PNs and ANs), however others may wish to and would therefore be interested to see this.

It would also have been good to have seen the conversion at different levels of NO and NO2 in the initial sample, corresponding to what would be the case in ambient measurements. Would the authors expect any difference in behaviour of the inlet with different NOx levels (e.g. any NO to NO2 conversion)?

Having read the title of the manuscript, I did find myself disappointed at the lack of results shown from the TD-LIF instrument. In reality this seems to be a paper that describes work done on the TD-CRDS system with a single set of results of HNO3 dissociation from the TD-LIF shown as an example of how two apparently similar systems can behave differently. This is fine and doesn't particularly detract from the results, however maybe the paper title is a bit misleading?

I found the section on ammonium nitrate conversion very interesting. It has often been a question when looking at total NOy measurements by some form of conversion to NO or NO2 as to how much nitrate aerosol is converted and this work goes some way to answer this. I wonder if a recommendation from the work could be that steps should be

taken to remove aerosols from the system (e.g. using a cyclone or something similar) if purely gas phase NOy is of interest? Maybe this could also be run sequentially to give an actual measurement of particulate nitrate? Could the authors make any comment on how other particulate nitrate (e.g. organic nitrate) would behave in the system? I would imagine there are situations when organic nitrates could make up a large proportion of the total particulate nitrate and so how these are converted would be of great interest.

Could the authors include a summary plot showing the thermograms of all the different species? Maybe this would look too busy but it would help the reader to understand the temperature that different conversions and interferences occur.

Minor corrections:

P2 L20: the authors could also include mention of GC-MS to measure organic nitrates (e.g. Worton et al., Atmos. Env. 2010) in their list of NOy techniques.

P4 L19: Are there any examples of using TD followed by photolytic conversion / chemi-luminescence to detect the NO2?

P5 L28: How are the know concentrations of NO2 produced (e.g. GPT, standard bottle with dilution)??

Figure 4: It would be better if these were plotted as a % conversion to NO2 as is done for all the other figures.

---

## Referee Comment (RC3) · Anonymous Referee #3 · 19 Jan 2017

This manuscript reports some tests to characterize the thermal dissociation technique for the detection of reactive nitrogen species using CRDS and LIF detectors. The manuscript is generally well written, and the main results confirm what already known from previous publications. On the other hand, there are some technical details and insights that can be of interest for more selective and precise speciated NOy measurements. I think that it fits with AMT scopes and I recommend publication, after the authors address the following questions and comments.

Main comments and questions:

Page 11, lines 4-13: Placing a stainless steel valve in front of the oven is something that I would avoid working with HNO3 that may be efficiently loss, even if the authors mention a test with the valve fully opened to check if the conversion of HNO3 changes when it is removed.

Page 12, line 3: Authors tested the effect of RH on the thermograph shape of the HNO3 with a test at 0% and another at 66%. Since in several sites the RH goes up to 90%, it would be worth to have one more point at high RH.

Page 15, lines 7-19: In this paragraph even if not clearly reported, it is implied that the thermal conversion of NH4NO3, reported also in figure 7, is a two step conversion: first from NH4NO3 to HNO3 and then from HNO3 to NO2, since the CRDS measures NO2. In this case it would be important the residence time to allow the double thermal dissociation in the heated tube, but this is not mentioned nor explored.

Page 15, lines 17-19 and figure 7: Here it is reported that the thermograph of NH4NO3 agrees with that of HNO3 reported in fig. 2. In fig. 2 are showed 4 thermographs of HNO3, but, to me, none of them are the same reported in figure 7.

Page 16, line 18: The NH3 conversion is unimportant for all the TD-LIFs, since all of them measure directly NO2: so I would generalize this conclusion to al the TD-LIFs and not only to the Berkeley TD-LIF.

Page 18, lines 22-23: This statement is not correct: 1) there are several campaigns where TD-NO2 were used during nighttime (i.e. BEACHON-RoMBAS, see Fry et al, 2013; RONOCO, see Di Carlo et al., 2013). 2) There is at least one paper where is described that during nighttime the channel of the TD-LIF instrument that converts

total peroxy nitrate into NO2, converts also N2O5 (Di Carlo et al., 2013). In that paper is reported also the comparison of nighttime measured peroxy nitrate by TD-LIF with the N2O5 measured by CRDS, taking the advantage of a TD-LIF and a CRDS installed on the same aircraft. In that work it is also showed that the TD-LIF measurements of peroxy nitrated, during nighttime and at least in the RONOCO campaign, are dominated by N2O5.

**Minor comments and questions:**

Page 6, line 1: the inlet tube 0.39 cm ID. Seems too small, is it a typo or a conversion error from inch to cm?

Page 15, line 5: Cohen, 2016 is cited as reference here, but it is not reported in the reference list.

**Reference**

Di Carlo P., E. Aruffo, M. Busilacchio, F. Giammaria, C. Dari-Salisburgo, F. Biancofiore, G. Visconti, J. Lee, S. Moller, C. E. Reeves, S. Bauguitte, G. Forster, R. L. Jones, and B. Ouyang, Aircraft based four-channel thermal dissociation laser induced fluorescence instrument for sim-ultaneous measurements of NO2, total peroxy nitrate, total alkyl nitrate, and HNO3, Atmos. Meas. Tech., 6, 971–980, 2013.

Fry, J. L., D. C. Draper, K. J. Zarzana, P. Campuzano-Jost, D. A. Day, J. L. Jimenez, S. S. Brown, R. C. Cohen, L. Kaser, A. Hansel, L. Cappellin, T. Karl, A. Hodzic Roux, A. Turnipseed, C. Cantrell, B. L. Lefer, N. Grossberg, Observations of gas- and aerosol-phase organic nitrates at BEACHON-RoMBAS 2011, Atmos. Chem. Phys., 13, 8585–8605, 2013.

СЗ

---

## Short Comment (SC1) · 13 Feb 2017

Womack et al. evaluated the conversion of dinitrogen pentoxide, nitric acid, ammonia, and ammonium nitrate in heated quartz inlets in two thermal dissociation instruments, the NOAA TD-CRDS and the Berkeley TD-LIF. They characterized the conversion of nitric acid as functions of flow rate, set temperature, relative humidity, and in the presence of $O_3$, CO, propane, and a VOC mixture. The measurements are novel and of great interest to users of TD instruments, of which there are now a handful (Paul et al., 2009;Thieser et al., 2016;Sadanaga et al., 2016;Day et al., 2002;Wild et al., 2014), and

complements nicely the recent work by the Crowley group (Sobanski et al., 2016).

I read this paper with great interest since my group has worked on the measurement of nitrogen oxides by TD-CRDS for some time. There is a lot of interesting information in this paper which are quite useful.

Below I would like to pass on some notes that I made reading this and will hopefully help improve this manuscript in its final version.

Title. The title seems a bit broad given that not all of the major NOy species were tested (e.g., PAN was not). Also, since measurement accuracy was not actually stated (e.g.," the measurements of .... are accurate to +/-x%" or something to that effect) , perhaps the title should be "Evaluation of interferences of ..."?

pg 1, line 27. The paper that should be cited for detection of $ClNO_2$ by CRDS is (Thaler et al., 2011).

pg 3, line 27. TD-CIMS instruments do not quantify ANs. They are usually quantified by clustering reactions with iodide and do not utilize a TD inlet.

pg 7, line 3. Typo (Marrin)

pg 9, lines 21-22, and all figure captions. Please specify which instrument was used to monitor $NO_2$. It was not always obvious.

pg 10, line 27. Sobanski (2016) is not listed in the reference section.

pg 11, line 25. "The Berkeley group has found the $HNO_3$ conversion to be oven dependent even for identical pressure and flow conditions indicating some but not all ovens have impurities at the walls that effectively catalyze $HNO_3$ decomposition." This statement has major implications and should perhaps be featured more prominently (maybe repeated in the conclusion section). Can the authors speculate as to what these impurities might be? How permanent are these effects? Could they, for example, occur between inlet characterizations in the field and compromise results?

pg 15, line 20. NH4NO2 – typo

pg 18, line 27. Slusher et al. 2004 is not a suitable reference as CIMS quantifies PAN and N2O5 at different masses and no corrections are necessary.

Figure S7. Not sure what is meant by 0 nm sized particles – maybe it should be "no particles"?   References Day, D. A., Wooldridge, P. J., Dillon, M. B., Thornton, J. A., and Cohen, R. C.: A thermal dissociation laser-induced fluorescence instrument for in situ detection of NO2, peroxy nitrates, alkyl nitrates, and HNO3, J. Geophys. Res., 107, D6, 4046, 10.1029/2001JD000779, 2002. Paul, D., Furgeson, A., and Os-thoff, H. D.: Measurement of total alkyl and peroxy nitrates by thermal decomposition cavity ring-down spectroscopy, Rev. Sci. Instrum., 80, 114101, 10.1063/1.3258204 2009. Sadanaga, Y., Takaji, R., Ishiyama, A., Nakajima, K., Matsuki, A., and Bandow, H.: Thermal dissociation cavity attenuated phase shift spectroscopy for continuous measurement of total peroxy and organic nitrates in the clean atmosphere, Rev. Sci. Instrum., 87, 074102, 10.1063/1.4958167, 2016. Sobanski, N., Schuladen, J., Schus-ter, G., Lelieveld, J., and Crowley, J. N.: A five-channel cavity ring-down spectrometer for the detection of NO2, NO3, N2O5, total peroxy nitrates and total alkyl nitrates, At-mos. Meas. Tech., 9, 5103-5118, 10.5194/amt-9-5103-2016, 2016. Thaler, R. D., Mielke, L. H., and Osthoff, H. D.: Quantification of Nitryl Chloride at Part Per Trillion Mixing Ratios by Thermal Dissociation Cavity Ring-Down Spectroscopy, Anal. Chem., 83, 2761-2766, 10.1021/ac200055z, 2011. Thieser, J., Schuster, G., Schuladen, J., Phillips, G. J., Reiffs, A., Parchatka, U., Pöhler, D., Lelieveld, J., and Crowley, J. N.: A two-channel thermal dissociation cavity ring-down spectrometer for the detection of ambient NO2, RO2NO2 and RONO2, Atmos. Meas. Tech., 9, 553-576, 10.5194/amt-9-553-2016, 2016. Wild, R. J., Edwards, P. M., Dube, W. P., Baumann, K., Edgerton, E. S., Quinn, P. K., Roberts, J. M., Rollins, A. W., Veres, P. R., Warneke, C., Williams, E. J., Yuan, B., and Brown, S. S.: A Measurement of Total Reactive Nitrogen, NOy, together with NO2, NO, and O3 via Cavity Ring-down Spectroscopy, Environm. Sci. Technol., 48, 9609-9615, 10.1021/es501896w, 2014.

---

## Author Comment (AC3) · 23 Mar 2017

The comment was uploaded in the form of a supplement:
http://www.atmos-meas-tech-discuss.net/amt-2016-398/amt-2016-398-AC3-supplement.pdf

---

## Author Comment (AC4) · 23 Mar 2017

The comment was uploaded in the form of a supplement:
http://www.atmos-meas-tech-discuss.net/amt-2016-398/amt-2016-398-AC4-supplement.pdf

---

## Author Response (AR1)

The authors would like to thank the reviewers of the manuscript entitled "Evaluation of the accuracy of thermal dissociation CRDS and LIF techniques for atmospheric measurement of reactive nitrogen species" for their helpful comments and suggestions. Our responses are as follows. The reviewer comments are in italics, our responses are in regular font, and changes to the manuscript are in blue.

**Reviewer #1**

*P2L25 Methods that detect some individual components of $NO_y$ are listed. How were they selected? Why not include e.g. $NO_3$ or HONO etc*

We wanted to highlight the detection of the just the largest components of reactive nitrogen. However, based on the feedback from both Reviewer 1 and 2, we have expanded this section to include references to $NO_3$, HONO, and other detection techniques for organic nitrates. P2L28 now reads: "HONO has been detected by long path differential optical absorption spectroscopy (Perner and Platt, 1979) and $NO_3$ has been detected by CRDS (King et al., 2000)."

*P3L24 "For example, TD-LIF detects $NO_2$ at low pressure following thermal dissociation, which minimizes secondary recombination reactions of the dissociated radicals". Are TD-LIF instruments always operated with the oven at low pressure which would minimise the recombination by reduction of reaction time and rate coefficient?*

TD-LIF instruments are sometimes operated with the oven at low pressure, but not always. In this sentence, we meant that the $NO_2$ detection (i.e. in the optical cavity, not the oven) always happens at low pressure, but have updated it for clarity. P3L26 now reads: "For example, TD-LIF detects $NO_2$ at low pressure following thermal dissociation. Secondary recombination reactions of the dissociated radicals would thus be suppressed in the detection region, although the thermal dissociation inlet may be operated at either high or low pressures in these instruments. However, it is subject to interferences from ambient levels of NO and $NO_2$…"

*P3L29 "TD-CRDS is an absolute measurement....." Does TD-CRDS being based on a cross-section of $NO_2$ really make it absolute? As stated later in the manuscript, the effective optical path-length needs to be calibrated by adding known amounts of $NO_2$. Also, the TD-inlet is part of the instrument and its dissociation efficiency needs to be calibrated (the subject of this paper).*

The CRDS measurement does not require calibration of the instrument response; it relates the ringdown time directly to concentration through equation (2), in which the calibration is an absorption cross section, making it an absolute measurement. The instrument is periodically compared to an $NO_2$ standard, but remains absolute. While characterization of the effective optical cavity length, $R_L$, is required, it is not necessary to use $NO_2$ for this process. Any gas-phase species which absorbs at 405 nm would allow us to measure this, it just happens that $NO_2$ is the most convenient. As for the TD-inlet, as long as the temperature setpoint is set correctly, no calibrations are required since the conversion efficiency should be a constant. Standard additions of specific $NO_y$ components would then still be useful for validation of instrument performance. The reviewer raises a good point that perhaps a better word for these steps is "characterization", rather than "calibration". We have updated the lines in the experimental and discussion sections to reflect this. P5L29 now reads: "$\sigma/R_L$ is characterized regularly by filling the cavity with several different known $NO_2$ concentrations." P20L5 now reads: "TD ovens should be characterized with the appropriate reactive nitrogen compounds regularly at the oven set points using the oven residence time and gas pressure that will be used in ambient sampling."

*P4L3 "though this reaction rate depends on the TD inlet pressure and flow rate". What reaction rate is this referring to (NO + $O_3$ makes $NO_2$ but O + $NO_2$ or NO does not)?*

The Wooldridge *et al.* paper was referring to the $O_3$ + NO reaction (among others) when it said these interferences are subject to pressure, so this is the reaction we were referring to. For clarity, we have removed the phrase "(or the O atoms that form in $O_3$ pyrolysis)" from the manuscript to clarify that we

are referring to the O$_3$ + NO reaction. P4L5 now reads: "Likewise, ambient levels of O$_3$ in the sampled air may react in the oven with NO to form NO$_2$, resulting in a positive bias (Pérez et al., 2007), though this reaction rate depends on the TD inlet pressure and flow rate (Wooldridge et al., 2010)"

5    *P4L10 "Thieser et al. (2016) parameterized the bias in peroxyacetyl nitrate and 2-propyl nitrate detection in their inlet as a function of ambient NO and NO$_2$ concentrations, but noted that these parameterizations may vary for other PNs or ANs. These effects are generally considered minor compared to other uncertainties in the measurement". Is this true? In some TD-instruments, depending on operating temperature, the effects of radical recombination (RO$_2$ + NO$_2$) or oxidation of NO (RO$_2$ +*
10    *NO) can bias the detection of peroxy nitrates by factors of 2 or more and is likely the biggest source of uncertainty.*
Thanks to the reviewer for pointing out this sentence was unclear. It was intended to refer to all the interferences described in this paragraph, and it was also intended to refer to after the biases were removed. However, we don't wish to speak for the authors of those papers, and we have therefore deleted
15    that phrase.
*P6L11 Does the temperature probe measure the gas-temperature or the temperature of the inner wall of the quartz tube? Can there be different due to gradients from the centre of the tube to the wall?*
The temperature probe is mounted on the outside of the quartz. As we point out on P6L12, the effect is that the actual gas temperature is slightly higher than the temperature setpoint, however, we periodically
20    monitor the gas temperature by inserting a temperature probe into the oven, as seen in Fig. S1.

*P6L17 Does addition of 30 ppmv O$_3$ have any adverse effects? Can e.g. ozonolysis of biogenics take place in this volume? Might this form particles or radicals (Criegees) than can react with NO$_2$?*
No, we are not concerned with this. Ozonolysis is quite slow, typically on the order of $10^{-17}$ cm$^3$/molec/s.
25    Even though the ozone concentration may be high, the concentration of unsaturated hydrocarbons which may undergo ozonolysis is typically at the level of a few ppbv or less. We would therefore expect less than 0.5% conversion, or an interference on the order of ~20 pptv, which is below the detection limit.

*P8L7 "A custom-built iodide adduct chemical ionization mass spectrometer (Lee et al., 2014), described*
30    *in further detail in (Veres et al., 2015), was used to monitor the N$_2$O$_5$ and HNO$_3$ concentrations" How was the CIMS calibrated? How accurate are the concentration measurements?*
As detailed in the paragraph starting on P14L19, the CIMS was calibrated for HNO$_3$ using a permeation tube, but no calibration source was available for N$_2$O$_5$, so a relative measurement was obtained. The CIMS measurements are accurate to within 20-25%. P8L12 now reads: "This measurement has a
35    detection limit of 4 pptv and 70 pptv and error bars of 25% and 25% (3σ) for N$_2$O$_5$ and HNO$_3$, respectively."

*P8L25 What is the pressure in the Berkeley oven? What are the standard operating conditions for this instrument. I believe it has been operated using different pressure configurations.*
40    The Berkeley TD-LIF instrument had an oven operating at ambient pressure, as described in Day et al (2002). P8L27 now reads: "HNO$_3$ and *n*-propyl-nitrate samples were provided by permeation tubes similar to those described in Sect. 2.2, diluted in dry zero air, and passed through 20 cm heated length quartz ovens, held at ambient pressure, at a flow rate of 2 slpm."

45    *P8L16 The modelling of this system is not trivial. As the authors state, many rate constants have not been measured at the higher temperatures. Secondly, the authors do not consider surface catalysed thermal decomposition, which is important as the authors mention briefly later when discussing the low temperature NO$_y$ instruments with catalytic conversion at the metal surfaces. The wall losses of radicals is probably the biggest uncertainty and can only be assessed by variation of experimental parameters.*
50    *Thieser et al. 2016 showed that variation of the concentration of the organic nitrates they were using (and thus variation of the RO$_2$ concentration) affected the loss rate, which could then be explained using*

*a Langmuir-Hinshelwood type expression. Did you change the concentration (e.g. of HNO₃) significantly to see if the same wall loss rate constant was appropriate? Do you expect the rate constant for wall loss to be independent of temperature (effects of diffusion, turbulent mixing)?*

We did not change the concentration of $HNO_3$, because we were limited by the output of the permeation tube, which, at a flow rate of 1.9 slpm provided a maximum mixing ratio of ~5 ppbv. We used this maximum concentration because it is where we would expect the highest probability of recombination. While we expect the effect of recombination to be even lower at more diluted concentrations, we didn't explicitly test conversion efficiency as a function of $HNO_3$ concentration. We have inserted a line stating this caveat. P10L24 now reads: "No attempt was made to dilute the output of the $HNO_3$ permeation tube any further, as recombination effects would likely only be less important at lower starting $HNO_3$ concentrations."

*P9L23 was the HNO₃ input mixing ratio based on the permeation source or the CIMS signal?*

It was based on the output of the permeation source. The CIMS signal is also calibrated on the output of the permeation source, so to base the calculation on CIMS signal would introduce more error into the calculation.

*P9L31…"possible due to recombination reactions……." Which ones? Be specific.*

We have clarified that the recombination reaction is of OH and $NO_2$. P10L5 now reads: "The 0.5 slpm thermogram has a slightly lower maximum conversion efficiency (95%), possibly due to the recombination reaction of OH and $NO_2$ during the extended time in the cool down region prior to detection."

*P10L8 High and low pressure limits have been used to calculate the thermal dissociation rate constant for HNO₃. What value for Fc was used to calculate the rate coefficient at 500∘ C. Also, Glänzer and Troe did their study in Argon. Are the results applicable for air (what is the relative collisional stabilisation efficiency)?*

We used a value of $F_c = 0.6$, and have put a note in Table S1 stating this. Following the example of Day et al (2002), we didn't attempt to make any correction for air vs. argon. To our knowledge, no studies of the $HNO_3$ thermal decomposition were conducted in air. Wine et al (JCP 1979) measured the recombination reaction in Ar and $N_2$ and found that although there was a ~30% difference in the rate constant at room temperature (at ~20 Torr), that difference decreased to 10% at 350K. They did not do any experiments at higher temperatures or at ambient pressure. So given the basic nature of the kinetic model, we didn't attempt to make a corrective factor.

*P10 L13 "…the recombination rate for OH + NO₂ is quite low…" Be quantitative. What is the rate coefficient at this temperature and wat is the pseudo-first order rate coefficient for recombination for a given NO₂ level of e.g. 10 ppbv. This can then be compared to the wall loss rate coefficient.*

The rate constant at 650 °C is on the order of $3 \times 10^{-13}$ $cm^3$ $molec^{-1}$ $s^{-1}$. Therefore, the pseudo-first order rate coefficient for recombination, given an $NO_2$ level of 10 ppbv is ~0.075 $s^{-1}$, nearly three orders of magnitude lower than the wall loss rate coefficient of 46 $s^{-1}$. We have inserted a line which describes this. P10L21 now reads: …OH radicals are far more likely to be lost to the walls of the oven (at a diffusion-limited rate determined by Day et al. (2002) of ~46 $s^{-1}$ for 1/4" OD tubing, which is far higher than the pseudo-first order recombination rate coefficient of 0.075 $s^{-1}$ at $[NO_2]$ = 10 ppbv).

*P10L27 Did Sobanski et al. (2016) also present a decomposition efficiency for HNO₃? Are the results comparable? Note that Sobanski et al used a radical scavenger with a large surface to reduce radical recombination in the heated inlet.*

They presented a decomposition efficiency curve up to 350 °C, and found the $HNO_3$ conversion efficiency was close to 0% at these temperatures. This is consistent with our results, and we have inserted that reference into the manuscript. P10L29 now reads: This is in contrast to ANs and PNs, for which the

reaction of the dissociated peroxy and alkyl radicals with $NO_2$ is a significant interference (Thieser et al., 2016), but in good agreement with the $HNO_3$ results of Day et al. (2002) and Sobanski et al. (2016).

*P11L9 ".. the onset and final conversion of $HNO_3$ are not strongly sensitive to pressure". Is this because wall losses are invariant with pressure? is the conclusion that wall losses are so large that recombination never compete ? What if the sample contains not only $HNO_3$ but also $NO_2$ to increase the rate of re-formation of $HNO_3$? The authors should consider doing one such experiment to see if elevated $NO_2$ will influence the shape of the $HNO_3$ thermogram. The same applies to the $NH_3$ expeirments.*
Yes, that is our conclusion. If we calculate the pseudo-first order rate coefficient for recombination with $[NO_2]$ = 50 ppbv, which is much higher than typically observed in the field, that rate coefficient is still two orders of magnitude lower than the wall loss rate coefficient, and is still unlikely to compete. We agree that testing this experimentally would be an interesting direction to take these studies in the future, but such tests were not feasible at this time. However, the kinetic box model supports this hypothesis, and we have inserted a line indicating this. P10L26 now reads: "Similarly, increasing the starting $NO_2$ concentration, to mimic conditions in highly polluted environments, was not attempted in this set of experiments, but increasing the starting $NO_2$ concentration in the kinetic model up to 50 ppbv shows that there is no recombination expected even with elevated $NO_2$ in the oven."

*P11L26 This sentence implies that the modelling done in this study (which considers gas-phase processes only) is only a partial representation of the chemistry going on. As mentioned above, the sensitivity $HNO_3$ detection while adding $NO_2$ would have been useful to confirm that the simple model reproduces the thermograms for the right reasons.*
Please see our response to the previous comment.

*P12L11 The data shows that the VOCs added had no effect. Not surprising considering their bond-dissociation energies. It would have been more informative to have added VOCs that will decompose, especially organic nitrates as they result in more complex radical chemistry and NOx.*
The purpose of the VOC additions were to test the effect on the secondary chemistry of $HNO_3$ conversion in high VOC environments, such as the recent Uintah Basin Winter Ozone Studies (see Wild 2016), where thermal dissociation inlets were used to measure $NO_y$ and speciated $NO_y$. While the reviewer is correct that adding organic nitrates would also be interesting, the scope this paper was limited to the listed components of reactive nitrogen, $HNO_3$, $N_2O_5$ and ammonium nitrate, to test quantitative conversion at high temperature. However, the study of Thieser *et al*. nicely defines the effect of organic nitrates. We have inserted a line clarifying the intention of these experiments. P12L23 now reads: "Figure 5b shows the measured thermogram with the addition of ~50 ppbv VOCs (described in Sect. 2.2) with and without the addition of 90 ppbv $O_3$, as well as the addition of 5 ppmv of propane, to mimic conditions found in highly polluted wintertime atmospheres."

*P12L17 ".....The oven is set at sufficiently high temperatures to dissociate ANs and PNs back to $NO_2$ + the organic radical" Not true. At higher temperatures the $RO_2$ formed from thermal dissociation of PNs is unstable (see Thieser et al. 2016).*
The reviewer is correct, $RO_2$ is often unstable at high temperatures. We were intending to say that if some $RO_2$ formed via ozonolysis or other mechanism, it is unlikely that it would scavenge $NO_2$ to reform PNs, since PNs would immediately dissociation. It is true that if the $RO_2$ simply dissociated, then it wouldn't react with $NO_2$ at all. Therefore, we have inserted a line which states that the $RO_2$ molecule would be more likely to dissociate. P12L30 now reads: "Reactions of unsaturated hydrocarbons with O atoms or OH radicals tend to be rapid and would produce organic radicals, but these tend to be unstable, and any stable radicals would likely only react with $NO_2$ to form ANs or PNs. The oven is set at sufficiently high temperatures to dissociate ANs and PNs back to $NO_2$ + the organic radical."

*P12L27 "....The dominant reaction of $O_3$ in the model is the reaction with $NO_2$ to make $NO_3$...." Is this true? I would have thought the pyrolysis will dominate at high temperatures. Is the $O_3$ pyrolysis rate constant in the model correct? What is the O-to-$O_3$ ratio at thermal and kinetic equilibrium?*
We have changed the line to distinguish those reactions from the unimolecular dissociation reaction, which should dominate at higher temperatures. P13L9 now reads: "The dominant bimolecular reaction of $O_3$ in the model is the reaction with $NO_2$ to make $NO_3$, but since these reactions are occurring at high temperature, any $NO_3$ formed will immediately dissociate to $NO_2$ (see Sect. 3.2)."

*P12L30 The reaction between O and $NO_2$ does not form much $NO_3$ but mainly $NO + O_2$. This is especially true at high temperatures.*
Both reactions should be relevant, so we have added both to that line. P13L13 now reads: "Of the O atoms that are not lost to the walls, their primary reaction is also with $NO_2$ to form either $NO + O_2$ or $NO_3$ but NO should be converted back to $NO_2$ after the oven."

*P13L5 That a model with no surface-catalysed reactions cannot reproduce the effect of a surface catalyzed reaction is not surprising. Why do NOy instruments with e.g. gold-surfaces see decomposition at much lower temperature than needed to break the HO-NO2 bond and why do they add CO? It is more than "possible" that surface reactions play a role, it is rather clear.*
We have removed the word possible, and replaced it with the word "likely". P13L25 now reads: "It is likely that there is some surface reaction that affects the $HNO_3$ conversion in the presence of CO."
*P16L25 Are there any other reactions of $NH_2$ that should be considered. Could it react with NO or $NO_2$?*
Thanks for noticing this. Although the $NH_2 + NO$ and $NH_2 + NO_2$ reactions were included in the model, they were mistakenly omitted from table S1. They have now been included.

*P17L6. "However, even this rudimentary simulation predicts the general shape of the experimental data..." What aspect of the "general shape" does it reproduce? Perhaps you can be more concise here.*
We have changed the line to be more specific. P17L24 now reads: "This rudimentary simulation predicts the initial signal increase starting at 300 °C, though it has a maximum conversion efficiency of just under 2%, which is below that observed in the experiment."

*P18L3 "ambient levels of a group of representative VOCs". As already mentioned, addition of VOCs that are unstable at the inlet temperatures (organic nitrates) would have been more informative.*
Please see our response to the comment on line P12L11.

*P18L23 "...$N_2O_5$ is not typically considered in the TD-$NO_2$ instrument literature because the existing instruments have largely operated in the daytime..." Perhaps this statement is too general. Some instruments measure day and night and have considered effects of $N_2O_5$ thermal decomposition (e.g. Thieser et al. 2016)*
That is true, there are some groups which have used TD inlets at night. We have rewritten that paragraph to incorporate this. P19L5 now reads: "TD-$NO_y$ instruments often operate in the daytime when $N_2O_5$ is not a significant fraction of $NO_y$, though some groups have operated at night and have typically assumed complete conversion to $NO_2 + NO_3$ at the TD inlet setpoint for PNs (Di Carlo et al., 2013), and complete conversion to $2NO_2 + O$ at the setpoint for $HNO_3$ (Wild et al., 2014). These results confirm that there is approximately quantitative conversion at these setpoint, though there are slight deviations from 100% conversion near the PN setpoint. Therefore, care must be taken to select a setpoint carefully and ensure complete conversion at that temperature. However, this interference would only be significant during nighttime or during very cold weather sampling."

*P18L32 "...These results demonstrate that the volatile portion of the particulate nitrates will be driven into the gas phase at low oven temperatures.." Particulate nitrate is not only ammonium nitrate but has a large component of organic nitrates. At which efficiency will these be detected?*

While we agree that organic nitrates are of significant interest, ammonium nitrate, which is in equilibrium with $HNO_3$ in the atmosphere, was the target of this study. We have inserted a line clarifying that we are talking about particulate ammonium nitrate. We have also inserted a line saying that although it's possible that other organic nitrates would behave similarly, further experiments would be required to test this. P19L22 now reads: "These results demonstrate that the volatile portion of the particulate ammonium nitrates will be driven into the gas phase at low oven temperatures, consistent with Rollins et al. (2010), who used a denuder to remove gas phase nitrates and to detect aerosol organic nitrates in a 325 °C oven. Their results indicate it is likely that particulate organic nitrates would be converted to $NO_2$ with 100% efficiency in the NOAA TD-CRDS, but this result has not been explicitly tested here."

Technical / typographical
*P2L20 Techniques that detect the major individual components of NOy include detection....*
See response to P2L20 above

*P4L1/3 negative / positive artefact = negative / positive bias?*
We have changed the word artifact to bias on P4L5 and P4L7.

*P5L4 Inappropriate reference. Fuchs et al were not the first to use CRDS for atmospheric trace gases as this implies.*
The intent was to cite a representative reference, not the first reference. We have added a reference at P5L4 to O'Keefe et al (1988), who were the first that we are aware of to measure ambient $NO_2$ with CRDS.

*P5L18 "light decays" ?*
We have changed "light decays" to "light decay profiles", for clarity. P5L18 now reads: "The measured light decay profiles are summed and fit at a 1 Hz repetition rate to yield the ringdown time τ."

*P5L23 "known NO2 concentrations". How were they determined? Was an "absolute method" used to measure the NO2 concentration?*
The known samples of $NO_2$ were obtained by reacting known amounts of $O_3$ from an ozone generator with an excess of NO. The ozone generator is a commercial ThermoScientific 49i, which measures $O_3$ by UV absorption. This is a technique that has been used by our group, and was described in section 2.2 of Washenfelder et al, EST 2011. We have inserted a new phrase to clarify this. P5L29 now reads: "$\sigma/R_L$ is characterized regularly by filling the cavity with several different known $NO_2$ concentrations (obtained by reacting the output of an $O_3$ standard source with excess NO) and calculating the slope of the measured optical extinction vs $[NO_2]$ as described in Washenfelder et al. (2011)"

*P7L22 "bubbling the dilution zero air through a water bubbler...."*
We have changed "bubbling" to "passing". P7L24 now reads: "Water was added by passing the dilution zero air through a water bubbler prior to mixing with the $HNO_3$ sample."

*P8L31 and at several other places in the manuscript. "rate laws" is the wring term. You refer to "rate constants" or "rate expressions".*
These have all been changed to rate expressions (throughout manuscript).

*References: Several have capitalsed manuscript titles. Nikitas et al spelling of Detector.*
Thank you for noticing. We have fixed these typos.

*Figure captions:*
*Figure 1: "Instrument schematic of the TD-CRDS instrument" "cool down" (sometimes cooldown" maybe you can find a better expression than "cool down region".*

We have changed this to "cooling region" in the Fig 1 and caption, which is consistent with what Day *et al.* (2002) and Wild *et al.* (2014) have called it.

*Figure 3: small dashed line = short dashed line ? Figure 4 "physical oven" ?? Figure 7 "in solid circles" = "as solid circles" ? Figure 8 "shown as red circles" Figure 9 delete preferentially.*
We have made these changes.

*Supplementary Info: Caption to Fig. S4. "...but if allowed to recombine, only ∼40% will be allowed to recombine, but that nearly all o atoms ........." Not clear what is meant here. Rewrite.*
This was a typo/error. We meant to say that only 40% will remain as $O_2$ + O. Figure S4's caption now reads: "These results indicate that $O_3$ dissociates to form O at the entire temperature range relevant for AN and $HNO_3$ TD ovens, but that if allowed to recombine, only ~40% will remain as $O_2$ + O. If wall loss is permitted, nearly all O atoms would be lost to reactions with the wall."
*Table S1. Many/most of the reactions listed contribute little to the thermograms (e.g. does neglecting H + NO3 make any difference at all)? Please highlight those reactions that do have an influence (i.e. those that account for 90 % of the reactive flux). This would make the results of the modelling exercise more transparent. Please add (in a footnote) the original references used for the rate expressions. Just listing the NIST type label (e.g. 1986TSA) is not sufficient. What does JPL \*\* mean (HNO3 + OH reaction). Please mark those reactions for which experimental data in the range up to 700 celcius was NOT available and inducate which (if any) are estimated or theoretical.*
We have reorganized Table S1. It now includes the rate constant at the minimum and maximum temperature (298K and 950K). We are also now indicating the temperature range, and whether they were experimental or theoretical. References are now directly included. The JPL \*\* was supposed to be a footnote which was mistakenly omitted, but which is now included. Any reactions at lower than 700 °C were used because reproducible values at higher temperatures were not available.

**Reviewer #2**

*Could the authors explain why they did not carry out experiments using common atmospheric NOy constituents such as peroxy and alkyl nitrates? Clearly these are likely to fully dissociate at the highest temperatures and this has been shown in previous work on the TD-LIF instrument, however it would have been good to actually see this shown experimentally. A thermogram of n-propyl nitrate from the UC Berkeley TD-LIF instrument is shown but it would have been nice to have seen this for the TD-CRDS system as well (especially considering one of the conclusions from the work is that all ovens behave differently). I do not know if the authors intend to use their system to provide NOy class speciation in their atmospheric measurements (and hence need to know the thermograms for PNs and ANs), however others may wish to and would therefore be interested to see this.*
The reviewer is correct that it is well worth looking at the conversion efficiency of peroxy and alkyl nitrates in a TD-CRDS instrument. Although we do not plan to use the TD inlet at intermediate temperatures for NOy speciation, others may, and we direct interested readers to the excellent paper by Thieser et al. (2016), which covered this topic nicely. We restricted our attention to quantification of the unintended thermal dissociation of $HNO_3$ at temperatures commonly used to measure ANs, and thus did not attempt to re-measure the conversion efficiency of PNs and ANs.

*It would also have been good to have seen the conversion at different levels of NO and $NO_2$ in the initial sample, corresponding to what would be the case in ambient measurements. Would the authors expect any difference in behavior of the inlet with different NOx levels (e.g. any NO to $NO_2$ conversion)?*
We appreciate and share the reviewer's concern. Please see our response to Reviewer #1's comment on P11L9

*Having read the title of the manuscript, I did find myself disappointed at the lack of results shown from the TD-LIF instrument. In reality this seems to be a paper that describes work done on the TD-CRDS system with a single set of results of $HNO_3$ dissociation from the TD-LIF shown as an example of how two apparently similar systems can behave differently. This is fine and doesn't particularly detract from the results, however maybe the paper title is a bit misleading?*

We understand the reviewer's concern, and it is true that this manuscript largely focuses on CRDS, but we would still prefer to retain LIF in the title, as it encompasses the results obtained by two of the paper's coauthors.

*I found the section on ammonium nitrate conversion very interesting. It has often been a question when looking at total NOy measurements by some form of conversion to NO or $NO_2$ as to how much nitrate aerosol is converted and this work goes some way to answer this. I wonder if a recommendation from the work could be that steps should be taken to remove aerosols from the system (e.g. using a cyclone or something similar) if purely gas phase NOy is of interest? Maybe this could also be run sequentially to give an actual measurement of particulate nitrate? Could the authors make any comment on how other particulate nitrate (e.g. organic nitrate) would behave in the system? I would imagine there are situations when organic nitrates could make up a large proportion of the total particulate nitrate and so how these are converted would be of great interest.*

Thanks to the reviewer for this suggestion. An impactor or cyclone could be used to eliminate particulate ammonium nitrate, whereas a heated aerosol sampler could potentially be used to sample 100% of the particulate matter in aircraft studies. As for other particulate nitrates, it seems likely that organic nitrates would be driven into the gas-phase with the same efficiency in our oven, given the results of Rollins et al (2010), and therefore would undergo complete conversion, but further tests would be required to test this. We have included lines in the discussion section which address these two issues. P19L24 now reads: "Their results indicate it is likely that particulate organic nitrates would be converted to $NO_2$ with 100% efficiency in the NOAA TD-CRDS, but this result has not been explicitly tested here." Additionally, P20L1 now reads: "In future studies, a TD inlet that either effectively samples aerosol, or effectively excludes aerosol (such as a cyclone), or a combination of the two could be used to specifically measure aerosol nitrates, which may make up a substantial fraction of $NO_y$, particularly in polluted wintertime urban atmospheres."

*Could the authors include a summary plot showing the thermograms of all the different species? Maybe this would look too busy but it would help the reader to understand the temperature that different conversions and interferences occur.*

Yes, we can do this. We have put it in the supporting documents as Figure S8.

*Minor corrections:*

*P2 L20: the authors could also include mention of GC-MS to measure organic nitrates (e.g. Worton et al., Atmos. Env. 2010) in their list of NOy techniques.*

Thank you for this suggestion. We have included the Worton paper from 2008 which introduced this technique in the introduction. (P2L33)

*P4 L19: Are there any examples of using TD followed by photolytic conversion / chemiluminescence to detect the $NO_2$?*

As far as we know, the only example of TD-chemiluminescence is Perez et al (2007), which dissociated HONO to make NO. We are not aware of any groups using TD-chemiluminescence to detect $NO_2$ directly. Because the line referenced here was only discussing techniques which generate $NO_2$, we didn't include that reference here, but did include it later in that section.

*P5 L28: How are the know concentrations of $NO_2$ produced (e.g. GPT, standard bottle with dilution)??*

Please see our response to Reviewer #1's comment on P5L23

*Figure 4: It would be better if these were plotted as a % conversion to NO₂ as is done for all the other figures.*

We have changed figure 4 accordingly.

**Reviewer #3**

Main comments and questions:

*Page 11, lines 4-13: Placing a stainless steel valve in front of the oven is something that I would avoid working with HNO₃ that may be efficiently loss, even if the authors mention a test with the valve fully opened to check if the conversion of HNO₃ changes when it is removed.*

It is true that stainless steel should normally be avoided, but Teflon valves did not provide a stable enough pressure to allow us to use them. Therefore, we used a heated stainless steel valve with the minimum amount of surface area possible to avoid any losses. The thermograms were also run over the course of hours to days, so the HNO₃ should have had enough time to come to an equilibrium with the surface.

*Page 12, line 3: Authors tested the effect of RH on the thermograph shape of the HNO₃ with a test at 0% and another at 66%. Since in several sites the RH goes up to 90%, it would be worth to have one more point at high RH.*

Unfortunately, this would have proved to be technically difficult to achieve, so based on the lack of a difference at 66% RH, we decided not to pursue higher RHs. However, the reviewer is correct that there could possibly be a non-linear water effect that is only evident at very high RHs, so we have included a line which describes this caveat. P12L22 now reads: "Additionally, we did not test the conversion efficiency at very high RH levels, and it's possible there could be a non-linear effect of water."

*Page 15, lines 7-19: In this paragraph even if not clearly reported, it is implied that the thermal conversion of NH₄NO₃, reported also in figure 7, is a two step conversion: first from NH₄NO₃ to HNO₃ and then from HNO₃ to NO₂, since the CRDS measures NO₂. In this case it would be important the residence time to allow the double thermal dissociation in the heated tube, but this is not mentioned nor explored.*

We anticipate that the thermal dissociation of NH₄NO₃, which takes place at much lower temperature, is rapid. Indeed, our model suggests that the thermal dissociation rate of HNO₃ is also quite rapid, and that the residence time in the heater is required largely to effect the temperature rise in the gas sample and not to allow time for the decomposition reactions. Furthermore, the shape of the thermogram, with a plateau at high temperature matching that of HNO₃, together with the calibration against an NH4NO₃ source, indicates complete conversion. We added a line indicating this. P16L3 now reads: "The close agreement between the two thermograms demonstrates that the dissociation pathway is NH₄NO₃ → NH₃ + HNO₃, and that this reaction is rapid at the temperatures reached in the TD inlet."

*Page 15, lines 17-19 and figure 7: Here it is reported that the thermograph of NH₄NO₃ agrees with that of HNO₃ reported in fig. 2. In fig. 2 are showed 4 thermographs of HNO₃, but, to me, none of them are the same reported in figure 7.*

The black trace in figure 7 is the same the one shown in gold squares in figure 4. Thank you for pointing out that this was not clear. The figure caption has been updated as "The black solid line indicates the measured thermogram of gas-phase HNO₃ at a 1.9 slpm flow rate (from the gold squares trace in Fig. 2)."

*Page 16, line 18: The NH₃ conversion is unimportant for all the TD-LIFs, since all of them measure directly NO₂: so I would generalize this conclusion to all the TD-LIFs and not only to the Berkeley TD-LIF.*

We have made this change. P16L32 now reads: "The interference is only present when O₃ is added to the mixing volume, indicating that the conversion of NH₃ must be producing NO, rather than NO₂, and is subsequently unimportant to instruments that measure NO₂ only, such as TD-LIF instruments."

*Page 18, lines 22-23: This statement is not correct: 1) there are several campaigns where TD-NO₂ were used during nighttime (i.e. BEACHON-RoMBAS, see Fry et al, 2013; RONOCO, see Di Carlo et al., 2013). 2) There is at least one paper where is described that during nighttime the channel of the TD-LIF instrument that converts total peroxy nitrate into NO₂, converts also N₂O₅ (Di Carlo et al., 2013). In that paper is reported also the comparison of nighttime measured peroxy nitrate by TD-LIF with the N₂O₅ measured by CRDS, taking the advantage of a TD-LIF and a CRDS installed on the same aircraft. In that work it is also showed that the TD-LIF measurements of peroxy nitrated, during nighttime and at least in the RONOCO campaign, are dominated by N₂O₅.*

Although the design of TD-LIF instruments has traditionally been oriented toward understanding photochemical reaction products of reactive nitrogen, we agree that the original statement was too general. We have rewritten that paragraph to account for studies that did use TD inlets at night. P19L5 now reads: "TD-NO$_y$ instruments often operate in the daytime when N$_2$O$_5$ is not a significant fraction of NO$_y$, though some groups have operated at night and have typically assumed complete conversion to NO$_2$ + NO$_3$ at the TD inlet setpoint for PNs (Di Carlo et al., 2013), and complete conversion to 2NO$_2$ + O at the setpoint for HNO$_3$ (Wild et al., 2014). These results confirm that there is approximately quantitative conversion at these setpoint, though there are slight deviations from 100% conversion near the PN setpoint. Therefore, care must be taken to select a setpoint carefully and ensure complete conversion at that temperature. However, this interference would only be significant during nighttime or during very cold weather sampling."

*Minor comments and questions:*
*Page 6, line 1: the inlet tube 0.39 cm ID. Seems too small, is it a typo or a conversion error from inch to cm?*
We used ¼" tubing, which has an inner diameter of 5/32" = .39 cm

*Page 15, line 5: Cohen, 2016 is cited as reference here, but it is not reported in the reference list.*
Thank you for noticing this. Cohen 2016 was indeed missing. This has been fixed.

**Comments from Hans Ostoff**

*Title. The title seems a bit broad given that not all of the major NOy species were tested (e.g., PAN was not). Also, since measurement accuracy was not actually stated (e.g.,"the measurements of .... are accurate to +/-x%" or something to that effect), perhaps the title should be "Evaluation of interferences of ..."?*
We understand Prof. Ostoff's concern, however, because we tested a number of NOy species, and because we tested them at a wide range of setpoints, not just the ones where they were supposed to be detected (i.e. HNO$_3$ at the ANs setpoints) we felt that it would be best to state them generally, rather than listing all the species out. Additionally, since the goal was not just to characterize interferences from NH$_3$ and O$_3$ additions, but to characterize how effectively the TD inlets convert species such as HNO$_3$ and ammonium nitrate particles, we would like to retain the phrase measurement accuracy.

*pg 1, line 27. The paper that should be cited for detection of ClNO₂ by CRDS is (Thaler et al., 2011).*
Thanks for catching this. We have fixed the reference.

*pg 3, line 27. TD-CIMS instruments do not quantify ANs. They are usually quantified by clustering reactions with iodide and do not utilize a TD inlet.*
This is true. We should not have included ANs in the list of species TD-CIMS detects. We have removed ANs from that line.

*pg 7, line 3. Typo (Marrin)*
We have fixed this typo.

*pg 9, lines 21-22, and all figure captions. Please specify which instrument was used to monitor $NO_2$. It was not always obvious.*
We have clarified that in all cases except when we are discussing the Berkeley TD-LIF, the NOAA TD-CRDS instrument was measuring $NO_2$. (Throughout manuscript)

*pg 10, line 27. Sobanski (2016) is not listed in the reference section.*
The reference is now listed.

*pg 11, line 25. "The Berkeley group has found the $HNO_3$ conversion to be oven dependent even for identical pressure and flow conditions indicating some but not all ovens have impurities at the walls that effectively catalyze $HNO_3$ decomposition." This statement has major implications and should perhaps be featured more prominently (maybe repeated in the conclusion section). Can the authors speculate as to what these impurities might be? How permanent are these effects? Could they, for example, occur between inlet characterizations in the field and compromise results?*
Unfortunately, we can't say for certain what those uncertainties are, or how permanent they are. This is why it is important to discard any ovens with obvious problems, and characterize the ones we do use very well. We have included a line in the discussion which emphasizes this. P20L5 now reads: "Based on the results of this paper, we make the following three recommendations: *(1)* TD ovens should be characterized with the appropriate reactive nitrogen compounds regularly at the oven set points using the oven residence time and gas pressure that will be used in ambient sampling. This is especially important given the findings of the Berkeley group regarding impurities found in otherwise identical ovens, as discussed in Sect. 3.1."

*pg 15, line 20. NH4NO2 – typo*
Thank you for catching this, we have fixed the typo.

*pg 18, line 27. Slusher et al. 2004 is not a suitable reference as CIMS quantifies PAN and $N_2O_5$ at different masses and no corrections are necessary.*
We were trying to say that Slusher *et al* had considered the recombination of $NO_3 + NO_2$ after the heater as a possible interference. However, it is true that this was not clearly stated, so because that paragraph had already been rewritten (see our response to Reviewer #3's comment on page 18, lines 22-23), we simply removed that statement.

*Figure S7. Not sure what is meant by 0 nm sized particles – maybe it should be "no particles"?*
This is indeed confusing. The 0 nm refers to setting the DMA size (and voltage) to 0, to ensure that no particles get through. Of course, it is possible for a few very small particles to get through, which is why we wanted to test the throughput at this voltage setting. We have included a line in the figure caption that explains this more clearly: "Here, "0 nm" refers to setting the DMA voltage to 0, which nominally does not allow any particles through."

[revised manuscript text omitted]

**Figure 10. Measurement of HNO₃ and NH₃ conversion in ambient air at an inlet set temperature of 650 °C. The left panel shows measured conversion efficiencies for standard additions of HNO₃ and NH₃ to the NOAA TD-CRDS  inlet sampling ambient air in Boulder, CO on August 9, 2016. The right panels show time series of measured NO$_y$ during standard additions. The data is the difference between two NO$_y$ measurement channels, one with and one without the standard addition, to cancel the variation in ambient NO$_y$ during the tests.**